# Performance based sub-selection of CMIP6 models for impact assessments in Europe

Tamzin E. Palmer[1], Carol F. McSweeney[1], Ben B.B. Booth[1], Matthew D.K. Priestley[2], Paolo Davini[3], Lukas Brunner[4], Leonard Borchert[5], and Matthew. B. Menary[6]

[1]Met Office Hadley Centre, FitzRoy Rd, Exeter, Devon, EX1 3PB, UK
[2]College of Engineering, Mathematics and Physical Sciences, University of Exeter, Exeter, UK
[3]Consiglio Nazionale delle Ricerchere, Istituto di Scienze dell'Atmosfera e del Clima (CNR-ISAC), Torino, Italy
[4]Department of Meteorology and Geophysics, University of Vienna, Vienna, Austria
[5]Climate Statistics and Climate Extremes, Centre for Earth System Research and Sustainability (CEN), Universität Hamburg, Germany. Laboratoire de Météorologie Dynamique (LMD) at École Normale Supérieure (ENS), Paris, France
[6]Laboratoire de Météorologie Dynamique (LMD) at École Normale Supérieure (ENS), Paris, France

**Correspondence:** Tamzin Palmer (tamzin.palmer@metoffice.gov.uk)

**Abstract.** We have created a performance-based assessment of CMIP6 models for Europe that can be used to inform the sub-selection of models for this region. Our assessment covers criteria indicative of the ability of individual models to capture a range of large-scale processes that are important for the representation of present-day European climate. We use this study to provide examples of how this performance-based assessment may be applied to multi-model ensemble of CMIP6 models to
5    a) filter the ensemble for performance against these climatological/ processed-based criteria and b) create a smaller sub-set of models based on performance, that also maintains model diversity and the filtered projection range as far as possible. Filtering by excluding the least realistic models leads to higher sensitivity models remaining in the ensemble as an emergent consequence of the assessment. This results in both the 25th percentile and the median of the projected temperature range being shifted toward greater warming for the filtered set of models. We also weight the unfiltered ensemble against global trends.
10    In contrast this shifts the distribution towards less warming. This highlights a tension for regional model selection in terms of selection based on regional climate processes versus the global mean warming trend.

## 1   Applications and motivations for regional sub-selection

Global Climate models (GCMs) represent one of the key datasets to explore potential future climate impact, vulnerabilities and risks. However, not all GCMs are equally skilful in capturing the climate processes that drive climate variability and change,

particularly at regional scales (Eyring et al., 2019). There is a growing interest, therefore, in assessing models and selecting
them for their suitability, if they are to be used to underpin or inform decision making. Such assessments are time consuming,
often pulling on diverse strands of evidence across the important physical and dynamical processes, which will vary according to region, application and variable of interest. This assessment information is also not commonly available to the broader
public making or using climate projection information. In this study we illustrate how such an assessment can be made for the
Coupled Model Intercomparison Project 6 (CMIP6) generation models, for projections in European regions. This provides an
assessment of how well these current models are able to capture the important regional processes over Europe. This information
can either be used by those focusing on particular processes or as a combined assessment, to identify which subset of models
may be more able to capture the relevant drivers of European climate change.

Historically, the climate modelling community has been cautious about weighting or eliminating poorly performing members due to the difficulties of linking performance over the historical period with future projection plausibility defaulting to a
'one model, one vote' approach (e.g. Knutti, 2010; IPCC, 2007, 2013). Whetton et al. (2007) evaluate the link between model
performance in the historical period and model performance for future projections by investigating the model similarity in
patterns of the current climate and the inter-model similarity in regional patterns in response to CO2 forcing. They find that
similarity in current climate regional patterns of temperature, precipitation and MSLP from GCMs is related to similarity in
the patterns of change of these variables in the models.

In addition while global temperature biases in the historical record are not correlated with future projected warming (e.g.,
Flato et al., 2013). This is not the case regionally for Europe, where biases in the summer temperatures have been found to be
important for constraining future projections (Selten et al., 2020). In addition projections of the Artic sea ice extent have also
been linked to historical temperature biases (Knutti et al., 2017). An increasing body of literature does link short comings in
the ability of a model to realistically represent an observed baseline to being an indicator that the models' future projections are
less reliable (e.g., Whetton et al., 2007; Overland et al., 2011; Lutz et al., 2016; Jin et al., 2020; Chen et al., 2022; Ruane and
McDermid, 2017). Regional model sub-selection is guided by a range of choices and there is always an element of subjectivity
in terms of how the criteria are determined. For example, if a model performs well for a particular target variable, but then
performs poorly in another season, variable, or location, this indicates that the regional climate processes are suspect (Whetton
et al., 2007; Overland et al., 2011).

To assess the model performance in terms of the regional climate processes, we firstly identify the key drivers of the European climate as our criteria. We then use these to assess the performance of the CMIP6 models across a range of variables. The
approach that we take one of elimination rather than selection and we do not recommend any individual model. Rather in our
examples of approach to sub-selection, we examine the impact on the projection range from the elimination of the models that
perform relatively poorly in these key criteria.

While there are strong arguments for filtering the ensembles for regional applications, the practical implementation requires us to navigate several challenges, such as how to select appropriate criteria, where the appropriate thresholds should lie for 'acceptable' vs 'unacceptable' models, and how to deal with models that perform well against some criteria but poorly against others. This inevitably introduces a degree of subjectivity in both the selection of the qualifying criteria and deciding the appropriate thresholds. For example, assessments of future changes in winter-time extreme rainfall in northern Europe are likely to emphasise the ability of simulations to capture the observed storm track position, whereas those assessments looking at summertime heat waves in central Europe may place more emphasis on ability of models to adequately represent summer blocking and land-atmosphere interaction processes. Advances in model development have led to significant improvements in the realism of regional processes, with incremental improvements in a number of long-standing biases and key processes (Bock et al., 2020).

Assessments and sub-selection of GCMs for regional applications have been implemented for CMIP6 using metric based approaches (e.g. Zhang et al., 2022; Shiogama et al., 2021). These studies aim to score or rank models for a particular region (Shiogama et al., 2021) or a range of regions based on a number of metrics (Zhang et al., 2022). Other regional approaches may weight GCMs based on regional performance against a range of metrics (e.g. Brunner et al., 2019). Weighting models regionally based on a range of metrics may produce mixed results however, and not always improve the ensemble mean bias. Assessments that are based on process-based analysis that emphasise region-specific process may produce better results (MS and JA, 2019). The Inter-Sectoral Impact Model Intercomparison Project (ISIMIP), aims to collate climate impact data that is consistent for both global and regional scales and across different sectors (Rosenzweig et al., 2017; Lange and Büchner, 2021). These studies use a limited number of GCMs (from CMIP5 and CMIP6) that are largely selected based on the availability of daily data for the required variables (Hempel et al., 2013). There have been concerns however, that the 4 GCMs used from CMIP5 in ISIMIP2b maybe unable represent the full range of uncertainty for future climate projections, especially for precipitation (McSweeney and Jones, 2016; Ito et al., 2020).

In this paper, we illustrate how current climate models can be assessed against their ability to capture a broad range of large-scale climate processes important for the European climate in the recent historical period. The rationale for doing so, is that models which do not adequately represent processes known to be important in the historical period for Europe, will not provide useful projections of future changes in these processes.

A processed based assessment, such as this, has several useful potential applications:

1. More robust European climate projections. By excluding models with the least realistic representation of regional climate drivers, we ensure that European projections are based on only those which can adequately capture present day processes. These remaining models are better candidates for understanding downstream impacts, both because their model biases are likely to be reduced compared to models that are unable to represent key features of the climate in the historical period, and

because we can have more confidence that they can capture the regional processes relevant to future changes.

2. Assess whether process-based evaluation has impact on the range of expected future changes. Such an assessment provides an opportunity to explore whether there may be any relationship between the quality of regional process representation and the range of changes projected from these models.

3. As an aid to further model development. Identifying where individual climate models have problems with particular regional climate processes, can be used to inform the type of model processes where further model development would be beneficial, both for individual models and for GCMs in general.

4. Define a reduced set of more reliable climate projections to inform subsequent sub-selections. Several approaches make
use of small(er) subsets of simulations, for computational or practical reasons or to simplify climate projection information. A performance filtered subset ensemble represents an important starting point for such a selection and there are different approaches that may be used:

a. Sub-selection matrix: Sub-selection is often used to identify a simpler set of data that retains the characteristics of the
underlying range of projected changes. This might be motivated either by computation (or other practical) limitations on the number of models and /or climate realisations that can be used in a particular application. In the case of sub-selecting a GCM matric for downscaling, Regional Climate Models (RCMs) will inherit errors from GCM boundary conditions. Therefore selection of models based on their ability to reproduce regional boundary conditions, such as features of large scale circulation is desirable (MS and JA, 2019). Alternatively, it might be motivated by the desire to reduce the complexity, by sub-selecting from the multi-model ensemble to still represent the underlying distribution as far as possible. Here, there is a need to balance criteria on credibility, with criteria to ensure that the subset can capture the broader range of potential changes and consists of as many independent models as possible.

b. Selecting individual realisations for use as climate narratives: Individual realisations are often used to exemplify responses
in certain parts of potential climate projection space. For example, selecting realisations to represent what central estimates or worst-case estimates, of future changes might look like. Alternatively selecting realisations that can be used to illustrate changes by particular drivers (e.g. the impact of strong changes in the NAO van den Hurk et al., 2014) or dynamical drivers of regional changes (e.g. Shepherd, 2019, 2014; Zappa and Shepherd, 2017). Pre-filtered ensembles based on regional performance metrics help identify more credible realisations that could be used as climate narratives.

Here we demonstrate performance filtering for CMIP6 models, against a broad range of climate process-based criteria relevant to Europe. This filtered subset can be used as a starting point, by others, to inform a selection of climate simulations appropriate for their own applications. This could be used by either drawing on individual assessment criteria or, as we go on

to show here, the outcome of filtering on the full set of assessment criteria. In this paper, we illustrate the implication of this filtering for the range of expected changes over Europe (point 2, above) and work through an example of how this could be used in conjunction with model diversity criteria, to identify a smaller subset of realisations suitable for driving downstream impacts relevant modelling.

The selection of GCMs for a particular region is an opportunity to exclude models that are considered to be 'Inadequate' in terms of their ability to represent key drivers of the regional climate. This has been attempted in a number of studies (McSweeney et al., 2015; Lutz et al., 2016; Prein et al., 2019; Ruane and McDermid, 2017), but it is still a challenge in terms of how to identify which models are 'Inadequate' and how the decision to eliminate these models should be made, particularly if their removal results in a significantly reduced projection range. Where the removal of a model, that is not considered to be able to give meaningful or useful information about the present or future climate, reduces the range of projections, this needs to be carefully justified. In addition to classifying models as either adequate or 'Inadequate', we look to classify models in a more informative way, and provide further information about how each of the CMIP6 models may perform in terms of key processes that influence the climate in the main European regions. The assessment is broken down into a number of different criteria that are scored individually, providing information regarding how individual models perform for each of these.

We build on the approach developed in McSweeney et al. (2015, 2018) previously applied to CMIP5. In McSweeney et al. (2015, 2018), CMIP5 models were assessed on a range of regional criteria, including the circulation climatology, distribution of daily storm track position, the annual cycle of local precipitation and temperature in European sub-regions. These characteristics were assessed using a qualitative framework for flagging poorly performing models as 'implausible', 'significantly biased' or 'biased'. This performance information was subsequently used together with information about projection spread (McSweeney et al., 2015) or model inter-dependencies (McSweeney et al., 2018) to arrive at sub-sets of the required size.

Many of the individual models and higher resolution model versions in CMIP6 show significant improvements in common model biases compared to CMIP5 (Bock et al., 2020). There are also a number of assessments in the literature that show an improvement in many of the processes that are key drivers of the climate for Europe e.g., Storm Tracks (Priestley et al., 2020, 2023), Blocking frequency (Davini and d'Andrea, 2020) and North Atlantic (NA) Subpolar Gyre (SPG) sea surface temperature (SST) (Borchert et al., 2021b). We draw on these analyses already in the literature to assess these large-scale processes for the European region, along with the assessment of features such as large scale circulation patterns, precipitation annual cycle and surface temperature biases using the method of McSweeney et al. (2015). Additionally, we look to classify models in a more informative way than simply keep or reject for sub-selection, to provide further information about how that model may perform in terms of key processes that influence the climate in a particular European region. Finally we note that our assessment is based solely on process-based criteria and does not use any regional or global warming trends, which separates it from many recent global assessments of CMIP6 (Tokarska et al., 2020; Brunner et al., 2020b).

In the following section we describe each of the criteria that have been selected along with their relevance for the European climate. We then define how each of the classifications that we use for the criteria are defined. In section 3 we present the methodology along with examples of how individual criteria have been assessed. In section 4 we examine the impact of filtering out models that fail to reproduce key processes on the projected range. We then use these performance filtered models to create a smaller sub-selection that also considers model diversity and maintains the projected range of the filtered models as far as possible. In sections 5 and 6 respectively, we discuss these results and present our conclusions.

## 2 Performance assessment for Europe

### 2.1 Criteria

#### 2.1.1 Atmospheric criteria

The near surface temperature and precipitation are key variables for future climate and are of primary consideration in impact studies, especially in terms of future hydrology considerations (e.g. White et al., 2011; McDermid et al., 2014; Ruane et al., 2014). They have been considered key variables in previous sub-sampling approaches (e.g. Ruane and McDermid, 2017; McSweeney et al., 2015).

A number of previous studies have considered the importance of capturing the main synoptic features and large-scale atmospheric circulation patterns (e.g. McSweeney et al., 2012, 2015; Prein et al., 2019) as a key criteria for GCM sub-setting. For northern Europe in particular large scale weather patterns and the passage of weather systems that make up the North Atlantic (NA) storm track dominate the climate, especially in the winter. Extratropical cyclones are the dominant weather type in mid-latitudes where they can have a significant impact due to associated extreme precipitation and windspeeds (Browning, 2004; Priestley et al., 2020). They have an important role in the general circulation in the poleward transport of heat, moisture and momentum (Kaspi and Schneider, 2013) and in maintaining the latitude westerly flow. In the winter (DJF) many GCMs have a southern bias in the peak storm track density with the prevailing winds too zonal resulting in higher than observed windspeeds across central Europe (Priestley et al., 2020; Zappa et al., 2013). In the summer (JJA) the prevailing wind direction is more westerly and less strong, but still an important driver of weather systems and key for representing the climate. We assess the large-scale circulation by comparing a baseline climatology with the ERA5 data (e.g. 1995-2014), using a similar approach to McSweeney et al. (2015). We use the analysis of Priestley et al. (2020) to assess the NA storm track over Europe in individual CMIP6 models.

Blocking by high pressure weather systems is known to cause of periods of cold dry weather in the winter and summer heatwaves. Blocking is typically under-represented in GCMs and this is still the case large parts of Europe in CMIP6, although there has been some improvement in the bias in many CMIP6 models (Davini and d'Andrea, 2020; Schiemann et al., 2020). We use the results of the analysis carried out by Davini and d'Andrea (2020) to assess the performance of the CMIP6 models

based on RMSE, bias and correlation.

### 2.1.2 Ocean criteria

The literature indicates that there is a link between NA Sea Surface Temperature (SST) and variability in the European climate (e.g. Dong et al., 2013; Ossó et al., 2020; Carvalho-Oliveira et al., 2021; Börgel et al., 2022; Sutton and Dong, 2012; Booth et al., 2012; Borchert et al., 2021a). The link between NA SST and drivers of the European climate is complex and how the atmosphere and NA interact over different timescales has not been fully determined. Representation of the NA SSTs in GCMs has also been shown to be key for other features such as blocking frequency (Scaife et al., 2011; Keeley et al., 2012; Sutton and Dong, 2012), Storm Tracks (Priestley et al., 2023) and the NA jet stream (Simpson et al., 2018). GCMs commonly feature a cold bias to the south of Greenland Tsujino et al. (2020), which is associated with biases in the latitude of the North Atlantic storm track due to unrepresented latent heat fluxes Priestley et al. (2023)). This cold bias commonly causes the storm track to be situated too far south Athanasiadis et al. (2022). Removing this SST bias results in improvements in the latitude of the atmospheric circulation Keeley et al. (2012) and in the simulation of other atmospheric phenomena such as blocking Scaife et al. (2011)). If this link between NA SSTs and the European climate remains important in the future a 'Satisfactory' representation of NA SST is required for also predicting the future European climate (e.g. Gervais et al., 2019; Oudar et al., 2020). There also appears to be some improvement in the skill in representation of the decadal NA and Sub-polar gyre in particular in CMIP6 compared to CMIP5 (Borchert et al., 2021b), which may be a factor for improvements in the representation of storm tracks (Lee et al., 2018), and blocking frequency (Keeley et al., 2012) for the European region in CMIP6 models compared to CMIP5.

The AMOC also plays a significant role in the present and future European climate due to its role in the poleward transfer of heat and ocean circulation. It also impacts on the NA SST (Jackson et al., 2022; Zhang, 2008; Zhang et al., 2019; Yeager and Robson, 2017), thereby influencing the SST impact on European climate discussed above. The CMIP5 and CMIP6 ensemble both predict a reduction in the AMOC by the end of century for higher emissions pathways, (Menary et al., 2020; Bellomo et al., 2021). The AMOC model comparison with rapid data from the analysis of (Menary et al., 2020) is used to assess the AMOC the GCMs.

### 2.2 Classification definitions

The purpose of this assessment is to identify models within the multi-model ensemble that are less capable of reproducing the processes that are relevant for the regional European climate. In terms of assessing the plausibility and performance of climate models, a degree of subjectivity is inevitably involved. One approach is to assess and rank the performance of the models based on a number of purely numerical measures of model error (RMSE, bias, variance, correlation), this provides valuable and objective information about the relative performance of the models, but it does not assess what the implications of the errors are, in terms how they impact on the ability of the model to make a meaningful regional projection. An additional qualitative

element to the assessment can add value by interpreting how these errors impact on the overall performance of the model for the regional climate and helps to inform the question of why these errors may cause a model projection to be less reliable.

A mix of quantitative (RMSE, bias, variance, correlation) and qualitative (e.g., inspection of circulation wind patterns) have been used and the models graded for each criterion using a coloured flag system. Visual inspection allows us to understand the characteristic of the error and consider its impact on other aspects of the model.

The models are given a classification 'flag' for each of the criteria described in the previous section, creating a table or coloured map summarising the performance of each model. This approach has been chosen, as opposed to a more quantitative metric for the assessment, to indicate where model performance for a variable is an issue. Where the qualitative assessment has been applied the quantitative metrics have been used as a guide to sort the models into classifications and also to ensure consistency as far as possible. The full details of how this has been applied to each criterion are described in the appendices (two examples are also given in the following section). In our assessment, models are therefore grouped into classifications that we define in this section.

Red: 'Inadequate' in performance criterion and should therefore be excluded from the sub-sample.

Orange: "Unsatisfactory" substantial errors in remote regions where downstream effects could be expected to impact on the reliability of regional information and/or present in the local region of interest.

White: 'Satisfactory', model errors not widespread or not substantial in the local region of interest. Location of substantial remote errors are not known to have a downstream impact on the local region of interest. Captures the key characteristics of the criteria spatially or temporarily.

Grey: Data/ analysis not available.

## 3 Materials and Methods

### 3.1 Data sources

Details of models from CMIP6 multi-model ensemble (Eyring et al., 2016) that are included in this study can be viewed in table S1 in the supplementary information. We use a baseline period of 1995-2014 and the period: 2081-2100 (end of century) for future projections. These time periods have been selected for consistency with existing EUCP analyses (e.g., Brunner et al., 2020a). We use the SSP585 scenario for comparison as this is the scenario with the strongest climate signal. The model data for the large area averages (for comparison of temperature and precipitation changes) is regridded onto a 2.5° x 2.5° grid and land-sea mask applied as used in Brunner et al. (2020a) and Palmer et al. (2021), using a standard nearest neighbour interpola-

tion. The data was averaged spatially using a weighted area mean.

The ERA5 reanalysis data and E-OBS gridded observational dataset (to evaluate the precipitation annual cycle) were used to assess the model error. Monthly mean data is used for the assessment with the exception of the blocking frequency analysis which uses daily data fields. Details of how these assessments have been carried out for each of the criteria, are given in the appendices. Examples of the assessments are also shown in the following section (section 3.2) for large scale circulation and storm tracks.

We use the results of the assessment, as described in the previous sections and summarised for the CMIP6 models, where sufficient assessment information was available in Figure 5. We use only the first realisation for each of the models in this assessment and assume that this is generally representative of the model performance. We acknowledge however that there may be a role for internal variability that pushes a model across assessment classifications. The largest uncertainty due to internal variability of the diagnostics we use is likely to be from the historical trends (which are not part of the assessment but used in an illustrative capacity). Brunner et al. (2020) found that for the global case the spread in the temperature trend fields between ensembles members of one model can be in the same order of magnitude as the spread across the multi-model ensemble. For the temperature climatology, in turn, the spread between ensemble members of the same model is typically less than 10% of the multi-model spread. This gives some indication that we can expect there to be relatively low variation in the performance of the models across the climatology for temperature based on which member is used. For the AMOC, which is a significant contributor to regional and global climate variability, Menary et al. (2020) noted that links to North Atlantic SSTs were sensitive to the removal (or not) of forced variability, but individual model realisations were not systematically different.

A case study is made to assess the role of internal variability for large-scale circulation (in which we may expect larger variability across ensemble members, than for the temperature climatology) in the CanESM5 model across all 25 realisations. This can be viewed in the supplementary information (Figs. S5 and S6). This context suggests that the analysis presented in this paper, based on the first ensemble member, likely provides an indicative picture typical of the response across any wider initial condition ensemble. However, future assessments may want to look for individual ensemble members which may show weaker manifestations of particular biases, particularly where a model lies close to classification boundaries.

## 3.2 Assessment examples

In this section we show two examples of the assessment method for two of the criteria discussed in section 2. Examples for all the criteria are given in the appendices.

For the assessment of each criterion, we refer to the model RMSE, bias and in some cases correlation with the reanalysis (e.g., for the precipitation annual cycle, see appendix A1 for details), in addition to a qualitative assessment of the model climatology, in terms of how errors impact on the ability of the model to represent the regional climate. In the process of

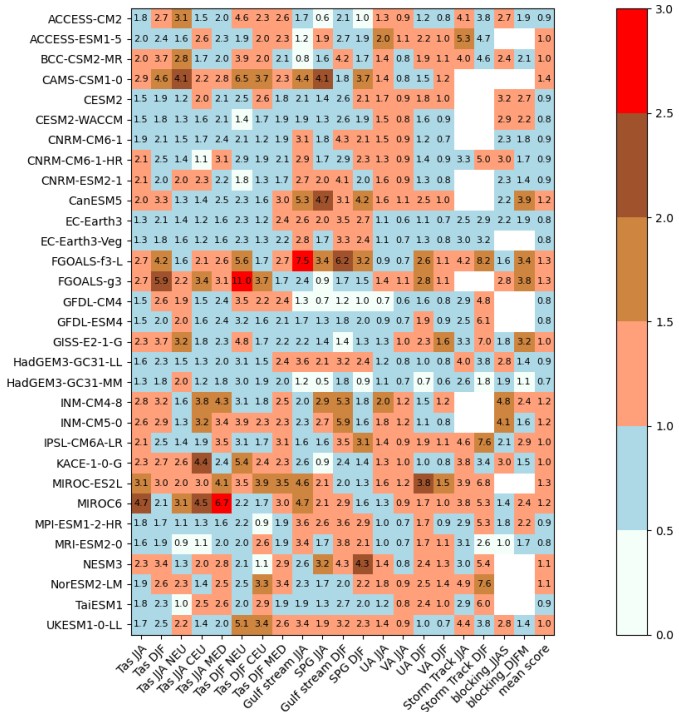

**Figure 1.** Summary of RMSE values for the large-scale assessment criteria and regional temperature. The regions are abbreviated as follows: northern Europe (NEU), central Europe (CEU), Mediterranean (MED). Tas: near surface air temperature, SPG: sub-polar gyre, UA , VA are the wind vectors at 850 hPa. The colour scale is determine by the ratio of the model RMSE to the ensemble mean RMSE. RMSE values are absolute, the mean score is the average of the relative error (normalised by the ensemble mean) across each of the criteria.

classifying the performance of the models, the qualitative interpretation of the errors has an element of subjectivity, as does the decision of where to place various thresholds for the quantitative measures. We aim to keep the assessment process as transparent as possible. In addition, it is important that while the qualitative assessment for an individual classification may occasionally differ to some degree from a purely quantitative approach, these decisions should not lead to the retention of

models with objectively larger errors in the sub-selection process. In the following section 3.2.1 (and in the appendices) we refer to both the fields of the model climatology and Fig. 1, which summaries the RMSE for each of the models.

### 3.2.1 Large scale circulation patterns

The large-scale seasonal circulation pattern was assessed for winter (DJF) and summer (JJA) based on the mean climatology at 850hPa for the baseline time period 1995-2014, the ERA 5 reanalysis was used for comparison (Fig. 2).

In DJF, European weather is dominated by the passage of weather systems that make up the NA storm track, the prevailing direction for these is from the south-west as can be seen in the climatology in ERA5 (Fig 2a). The model large-scale RMSE for the 850hPa wind vectors (e.g. Ashfaq et al., 2022; Chaudhuri et al., 2014), along with a qualitative assessment of the overall circulation pattern were used to assess the models for this criterion. Fig. 1 show that the wind vectors RMSE is less that of the multi-model mean for CNRM-CM6-1 and HadGEM-GC31-LL. Where the wind vector errors for a model is less than the multi-model mean the large-scale circulation is found to be reasonably well represented. Fig 2b) and c) show that these models capture the overall circulation pattern well and have relatively low windspeed biases. Where the models have a larger RMSE for wind vectors than the multi-model mean, the threshold for an 'Unsatisfactory' model requires some consideration. For these cases a qualitative approach is used understand how these errors may impact on the European climate and to guide where this threshold should lie.

The model with the largest errors of the 'Satisfactory' models is CESM2, with an area of positive bias over the UK, this model was still assessed as 'Satisfactory' however, due to the well define south-westerly wind patterns and good representation of the winds over most of the European land areas. The strength of the south-westerlies over the UK and Scandinavia is too weak in some of the models (e.g. IPSL-CM6A-LR Fig. 2f), along with the prevailing wind direction being too westerly. These models were flagged as 'Unsatisfactory'. These models feature a variety of structural biases, for example INM-CM4-8 which had a lower spatially averaged RMSE windspeed error, but lacked a clear representation of the south westerlies over northern Europe. The winds are too weak in these areas and there are areas of negative bias in the Mediterranean. This model was classified as 'Unsatisfactory' due to its lack of representation of the circulation pattern and general wind direction too westerly (Fig. 2g). This is also reflected in the wind vector errors (Fig. 1).

Models flagged 'Inadequate', have an almost entirely westerly (no south-westerlies) wind pattern and the wind speed errors over large parts of Europe are widespread and substantial (e.g., Fig. 2h-j, CanESM5, FGOALS-g3). These models nearly all have a large (positive) bias over European land regions (e.g. $> 6\text{ms}^{-1}$). MIROC-ES2L has the largest errors for the wind vectors in the ensemble for DJF (more than twice the ensemble mean error for UA), the errors do not follow the same pattern as the other 'Inadequate' models, with a large negative bias over most of Europe and an almost northerly wind direction in the NA (Fig.2i).

Circulation patterns are more westerly with weaker winds in the summer (JJA). These were assessed using the same approach for comparison as for winter circulation (Fig 3). Many CMIP6 models capture the general pattern well (e.g HadGEM-GC31-

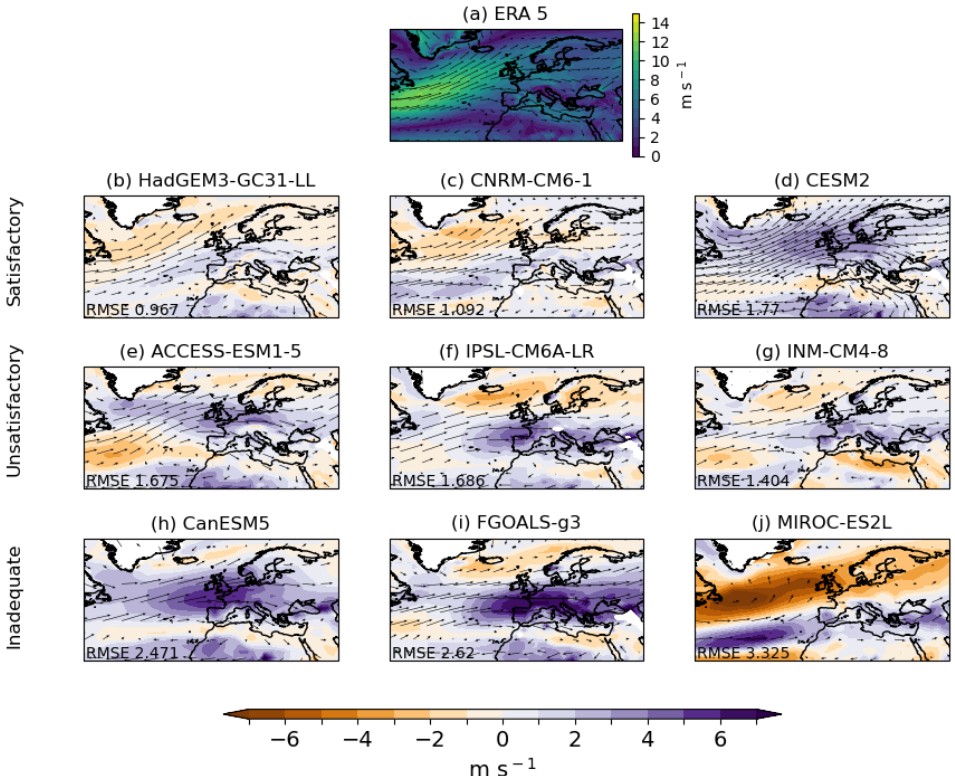

**Figure 2.** Examples of DJF circulation (850hPa) classifications for a sample of individual models. Top panel shows ERA5 climatology. Windspeed and direction are shown as a 20 year mean 1995 – 2014. Arrows show direction (absolute) of windspeed (scaled by windspeed) for climatology across all panels. The shading for the 3 panels shows the difference in windspeed between the model and ERA climatology.

LL, GFDL-ESM4, Fig 3b and d). The UA, VA RMSE for both of these models is less than the ensemble mean RMSE. Again, where the models perform above the average for the multi-model ensemble the overall circulation pattern is well represented with relatively low windspeed bias. As with the case for the DJF circulation where the models have larger errors than the multi-model mean for JJA wind vectors, the threshold is to warrant a flag as 'Unsatisfactory' or 'Inadequate' as been determined alongside some qualitative interpretation of the model errors.

Some of the models had westerly patterns over the UK and central Europe that were too weak (e.g. MIROC6, INM-CM4-8, Fig. 3f and h), as a result there are larger errors in European land regions and these models were therefore classified as 'Un-

satisfactory' or in the case of INM-CM4-8 where these errors are more pronounced , 'Inadequate'. In the case of MIROC6 we
note that the magnitude of the UA and VA errors over the large-scale region assessed as a whole are on the borderline of the
threshold for 'Satisfactory' and 'Unsatisfactory' compared to the other models. It is the relatively weak circulation and low
bias in windspeed over the European land regions that is the reason for the 'Unsatisfactory' flag in this case (Fig. 3e).

The INM-CM4-8 (and to a similar extent the INM-CM5-0) model has some of the largest errors for the JJA wind vectors
in the multi-model ensemble. It is noted that these models are also flagged as 'Inadequate' for both severe JJA blocking errors
and severe errors in representing the annual precipitation cycle in central Europe. There are also issues with the temperature
bias in central Europe for this model (flagged 'Inadequate'). These severe errors in central Europe are likely to be related to
the representation of the large-scale circulation.

For NorESM2-LM and ACCESS-ESM1-5 (Fig 3i and j) the westerly pattern was too far north leading to a large area of
positive bias over northern Europe. These models have the largest RMSE for wind vectors in the multi-model ensemble along
with the INM-CM4-8 and INM-CM5-0 models and the largest RMSE for windspeed. The large region of substantial positive
bias over the NA and much of Europe indicates that this error is likely to impact on the JJA storm track over Europe for these
models. As the storm track assessment is available for both these models this can be confirmed to be the case. The storm track
RMSE is in the top 85th percentile for the models assessed for the storm tracks, (see the following section on the storm track
assessment) and Fig. 1 shows that these models have the largest errors for the JJA storm track in the ensemble.

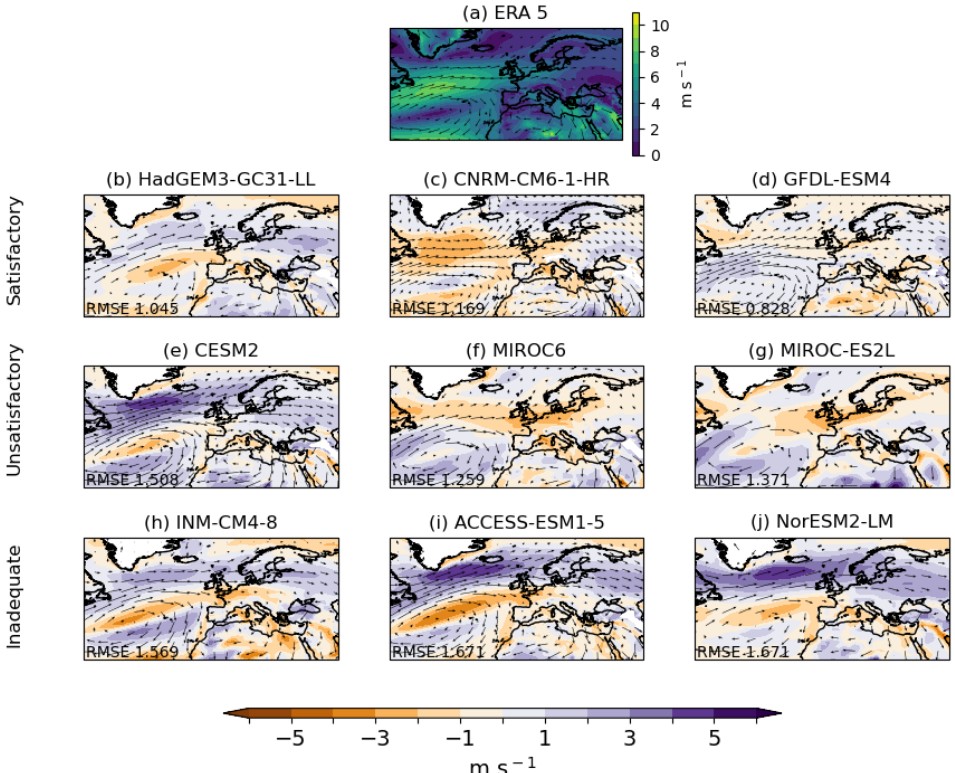

**Figure 3.** Examples of JJA circulation (850hPa) classifications for a sample of individual models. Top panel shows ERA5 climatology. Windspeed and direction are shown as a 20 year mean 1995 – 2014. Arrows show direction (absolute) and windspeed (scaled by windspeed) for climatology across all panels. The shading for the 3 panels show the difference in windspeed between the model and ERA climatology.

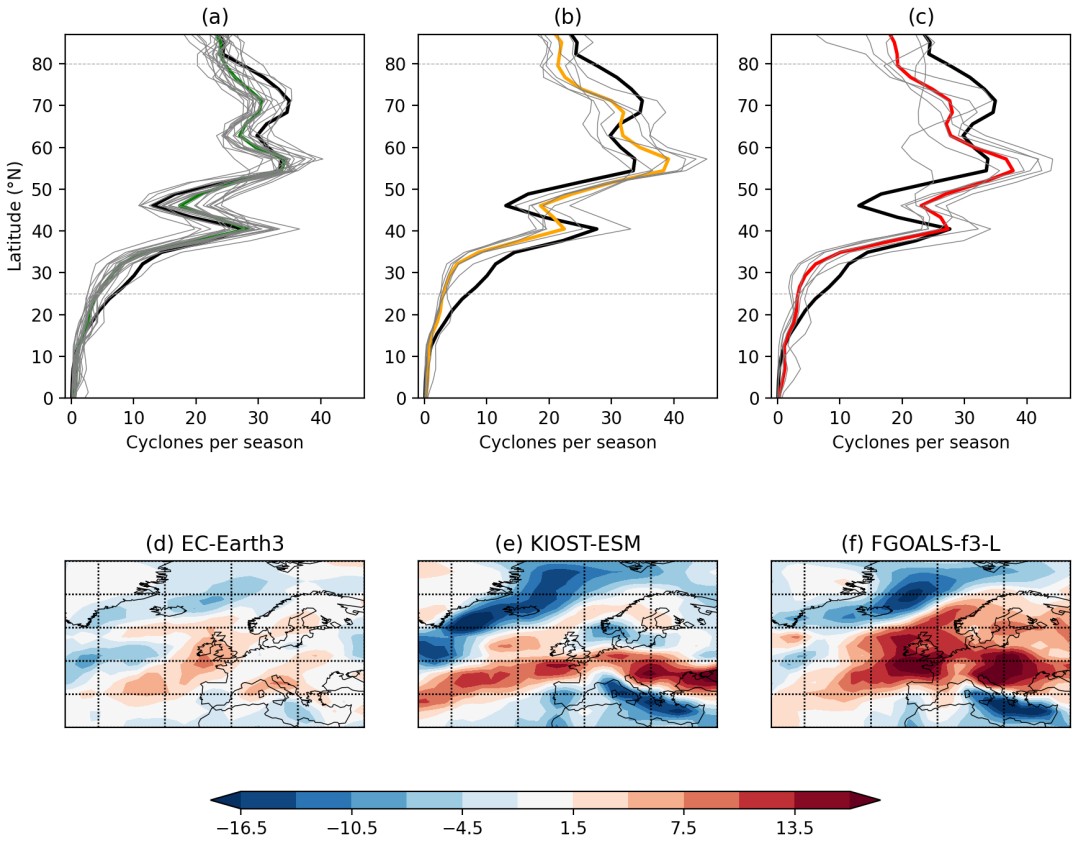

**Figure 4.** Examples of DJF storm track classifications. a), b) and c) shows the RMSE of the zonal mean track 20°W -20°E for individual models and the classification mean, for 'Satisfactory' (a), 'Unsatisfactory' (b) and 'Inadequate' (c). In (a–c) gray lines are individual models, solid-coloured lines are the group average, and black solid line is ERA5. Individual examples are shown in the lower panel for track density bias for 'Satisfactory'(d), 'Unsatisfactory' (e) and 'Inadequate' (f) models. Units of (d–f) are cyclones per season per 5 degree spherical cap.

### 3.2.2 Storm track large scale assessment

The track density is calculated using an objective cyclone tracking and identification method based on the 850 hPa relative vorticity (Hodges, 1994, 1995). The method and data are the same used in Priestley et al. (2020). The zonal mean of the model mean track density from 20°W-20°E was taken to get a profile of storm number by latitude. Then the RMSE was calculated of the models compared to the profile obtained from ERA5. The RMSE was calculated from 25-80°N.

The storm track has been assessed as a large scale feature using an assessment of the characteristic trimodal pattern (Fig 4) calculated as the zonal mean of the seasonal track density between 25°N-80°N and 20°W-20°E, compared to ERA5 reanalysis data. The baseline time period used for this assessment is 1979/80-2013 (as in Priestley et al. (2020)). The RMSE of zonal mean

track density from 20°W-20°E is used to initially sort the models into categories, while a hard cut off threshold was not applied for each category it was helpful to sort the models in < 65th RMSE percentile, 65th- 85th and > 85th percentile for RMSE. The different model groups were then inspected visually, it was found that although some of the models in the < 65th percentile had some significant biases the models in this group had clearly define peaks in their number of cyclones at the correct latitude and therefore captured the passage of storms across western and central Europe 'Satisfactorily' (Fig. 4a). This was not found to be the case for the models in the 65th - 85th RMSE percentile where there was a lack of a northern peak, this indicates a zonal bias in these models, which is a characteristic bias in GCMs (Fig. 4b). These models were classed as 'Unsatisfactory' although the errors were not large enough on visual inspection to class them as 'Inadequate', with the exception of MIROC-ES2L.

Models with >85th percentile RMSE failed to capture the tri-modal pattern, and had large biases in the number of cyclones at each of the peaks (Fig. 4c). In particular there was a lack of northern peak and an amplification of the errors in this group, with a large zonal bias in the track density. These models were considered unable to represent this feature and were flagged as 'Inadequate'. Examples of individual models for each of the groups are shown in Figure 4d-f. The RMSE values for each of the models in the multi-model ensemble are also shown in Fig. 1.

### 3.3 Weighting for performance against global trends and model independence with the climWIP method

We also compare our results with the Climate model Weighting by Independence and Performance (ClimWIP) method (Knutti et al., 2017; Lorenz et al., 2018; Brunner et al., 2019, 2020b; Merrifield et al., 2020) to assess differences between our process-based filtering and an assessment based on historical warming. ClimWIP combines model performance weighting based on one or more metrics with an assessment of model independence (i.e., overlaps in the models' source code or development history). Here we use an adaptation of the approach described in Brunner et al. (2020b) and publicly available via the ESMValTool (https://docs.esmvaltool.org/en/latest/recipes/recipe_climwip.html). Performance weights are calculated based on global temperature trends compared to ERA5 in the period 1980-2014. Independence weights are based on global model output fields for temperature and sea-level pressure which have been shown to reliably identify model dependencies (Brunner et al., 2020b; Merrifield et al., 2020). Here we use ClimWIP in two setups: one only based on performance weights, and one only based on independence weights as detailed later.

## 4 Results: Assessment and applications for sub-selection

### 4.1 Assessment Table

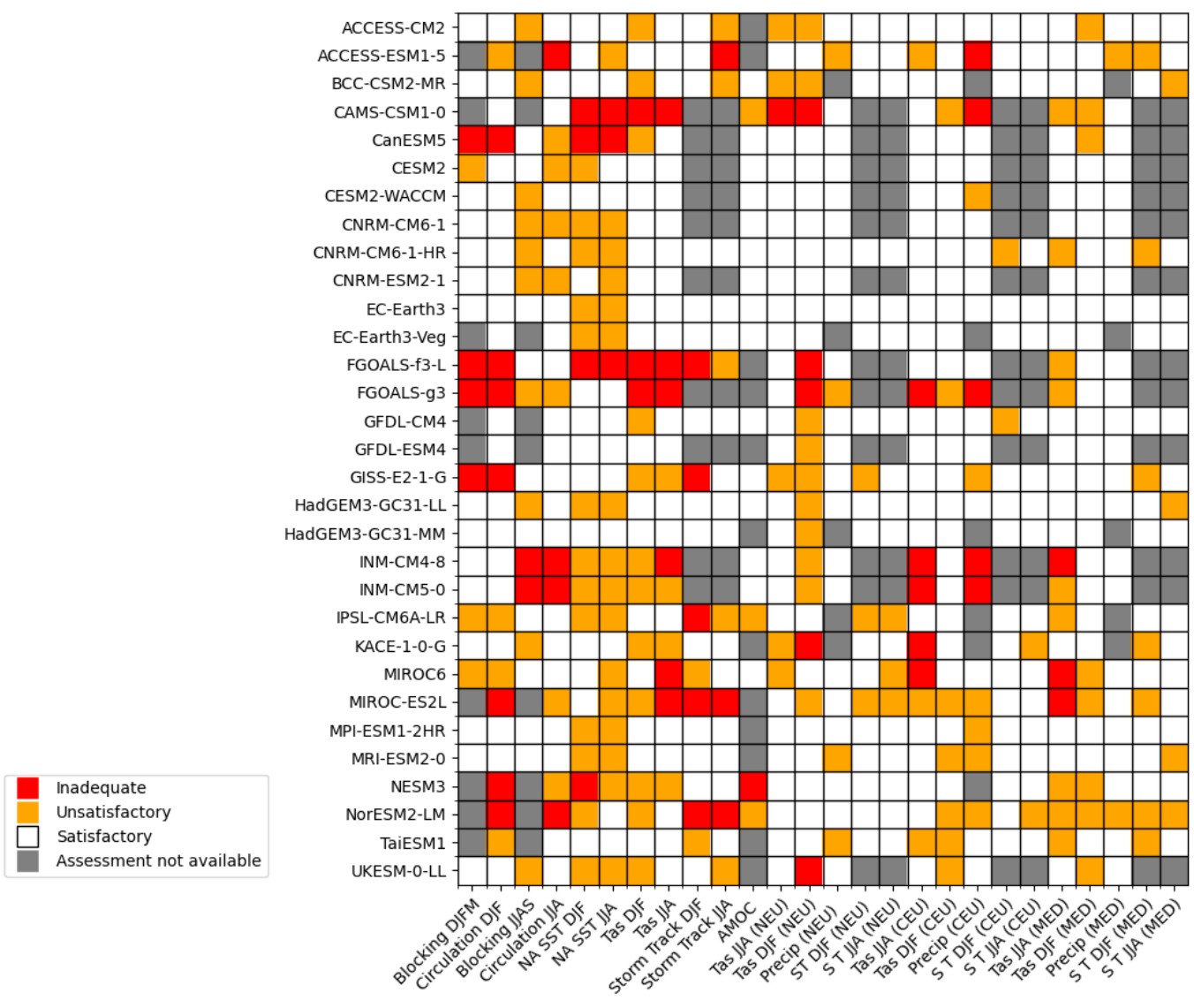

**Figure 5.** Model assessment summary for qualitative European criteria. Assessment criteria for the large scale are as follows. Blocking: Blocking frequency, Circulation: large scale circulation assessed by 850hPa windspeed and direction, NA SST: NA SST bias, Tas: surface air temperature bias at 2m, Storm Track: based on RMSE of the zonal mean track 20°W -20°E, AMOC: based on strength at 1000m at 26°N. Assessment criteria for the European regions are as follows. Tas: surface air temperature bias at 2m, Precip: Annual precipitation cycle, S T: storm track assessed as cyclones per season within European region.

The assessment for each of the CMIP6 models is collated into Fig.5 , with the classification for each of the criteria, where the relevant data and/or analysis are available. Fig. 5 creates a summary of each model's performance against a range of criteria, that are essential for a meaningful representation of the European climate. This summarises the skill, across a multi-model ensemble from CMIP6, of their ability to capture the key process for the European climate. The assessment criteria are divided into large-scale and regional assessments. The large-scale assessment criteria, such as large-scale circulation and blocking frequency, are criteria that have a pan European impact and are not specific to a particular region. The regional assessment criteria have been scored individually for each of the three main European regions used in the EUCP study and as defined as in Brunner et al. (2020a) and Gutiérrez et al. (2021). These are, Northern Europe (NEU), Western and Central Europe (CEU) and the Mediterranean (MED) (see Fig S1, in the supplementary information). We focus in the assessment on summer (JJA) and winter (DJF).

Some of the criteria were assessed both at the large scale and regionally. For example, it is useful to know if a model has a widespread temperature bias that extends over Europe and the NA, but it is also the case that some models have more localised temperature biases that affect individual regions. For the regional assessment where surface variable (e.g., precipitation, temperature) are assessed models were scored for their performance solely over the land regions.

The classifications in Fig. 5 can be applied to create a bespoke sub-set of CMIP6 models depending to on the motivation for sub-selecting. Here we have used the red classification of 'Inadequate' to indicate that a model should be removed, but it may be the case that a less strict approach to performance filtering is acceptable in some cases, than we have applied here. Likewise, it may be the case that an 'Unsatisfactory' (orange) flag in a certain criterion such as the regional precipitation may be particularly undesirable. In the following section we use the table to create two different sub-sets from the multi-model ensemble.

## 4.2 Excluding the models least representative of key regional processes

In this section we explore the implication of screening out poor models based on the process-based performance assessment alone, on the range of projected regional changes. The aim is to revisit the range of projected regional climate changes, excluding those shown to struggle representing regionally relevant processes. Note we do not include any criteria based on climate sensitivity or global temperature trends in this selection for this reason. These additional considerations and how they could be applied will be discussed along with the results.

For the sub-selection process, we refer back to the definition of the classifications in section 2.2. The 'Inadequate' category (shown as a red flag on Fig 5) is used to indicate that a model fails to represent a key feature of the regional climate and should be removed from the sub-selection. We also differentiate between large-scale criteria than can be expected to have pan European effects on the model performance (and may also be inherited from the GCM in case of down-scaling) and regional criteria, which may only be of concern in the local region. Here we consider the impact on the projection range of firstly excluding any model with one or more 'Inadequate' (red) flags for any of the large-scale criteria. We then go on to consider any further changes in the projected temperature range as a result of removing an remaining models with a regional 'Inadequate' flag.

Once all the models in Fig. 5 that have a red flag for the large-scale criteria are removed the following models remain in the sub-selection; ACCESS-CM2, BCC-CSM2-MR, CESM2, CESM2-WACCM, CNRM-CM6-1, CNRM-CM6-1-HR, CNRM-ESM2-1, EC-Earth3. EC-Earth3-Veg, GFDL-CM4, GFDL-ESM4, HadGEM3-GC31-LL, HadGEM-GC31-MM, MPI-ESM1-2-HR, MRI-ESM2-0, KACE-1-0-G, TaiEMS1, UKESM1-0-LL. This sub-selection from the qualitative assessment can also be compared to the RMSE values in Fig.1. If we look at the scores for the large-scale criteria (all categories in Fig 1, excluding regional temperature), it can be seen that the excluded models include all those with a RMSE more than 1.5 times the ensemble mean in at least one of the large scale categories. It is also the case that for the retained models that the RMSE does not exceed 1.5 times the multi-model ensemble mean for any large-scale category. The retained models also perform better than, or at least equal to the ensemble mean across all the categories. This indicates that in our application of the assessment objectively poorer models have been removed (in terms of large-scale performance) and those with objectively smaller errors have been retained.

Fig.6 shows the difference in the projected temperature range for the large-scale process-based filtered sub-set and the un-filtered multi-model ensemble. The difference in DJF is small (Fig. 6b), however in JJA the lower part of the range is reduced, and the upper part is shifted upwards (Fig. 6a). This shift in the projection range indicates that more of the higher sensitivity models are retained by filtering using process-based performance criteria.

In the second stage of filtering, we again refer to the regional criteria in the assessment table. There are 'Inadequate' (red) flags for regional precipitation (in central Europe) and for regional temperature in a few of the models (Fig.5). The models with

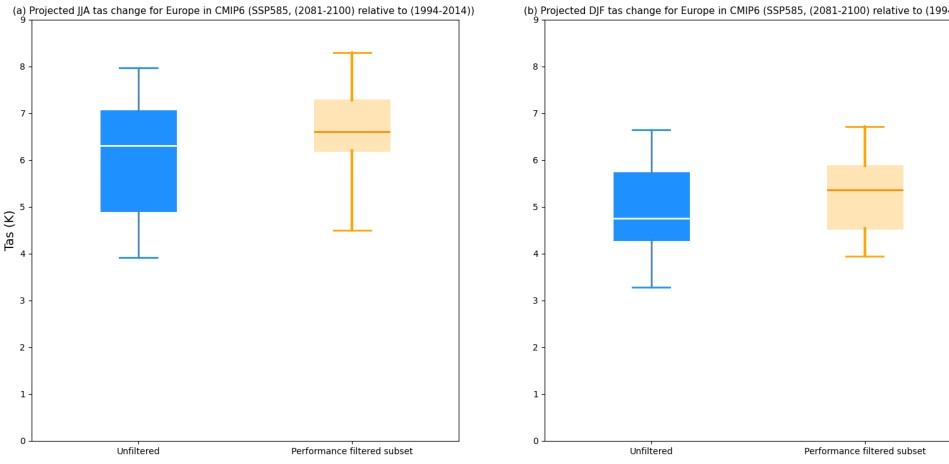

**Figure 6.** a) Projected range of JJA temperature change for Europe in CMIP6 (SSP585, (2081-2100) relative to (1994-2014)) for the raw unweighted multi-model ensemble and the large-scale performance filtered subset. Boxes show 25th to 75th percentile. Whiskers show the 5th and 95th percentile. b) As for a) but for DJF.

an 'Inadequate' classification for precipitation (INM-CM4-8, INM-CM5-0, ACCESS-ESM1-5 and FGOALS-g3), already have
at least one 'Inadequate' flag for the large-scale atmospheric criteria. Therefore, these models have already been removed from the performance filtered sub-set. The KACE-1-0-G model has two 'Inadequate' flags for regional temperature in two regions NEU and CEU. The UKESM1-0-LL model has a single 'Inadequate' (red) flag for temperature DJF (NEU). Fig. 1 shows these temperature errors in both models to be relatively large compared to the multi-model ensemble mean RMSE. In addition, the UKESM1-0-LL model has a relatively large DJF temperature error for CEU, indicating that this temperature bias extends over
two of the European land regions. These errors that are limited to specific regions may be considered acceptable for some applications, so may not necessarily always be a reason to exclude a model from a sub-selection. Here we filter the sub-set further by removing these models. Referring to Fig. 1, we can confirm that our excluded models include only those with a relatively large RMSE (1.5 times the ensemble mean) in at least one of the criteria. Also, that the eliminated models on average across the criteria have a relative error at least equal to or larger than the ensemble mean (Fig. 1). Therefore, it is again the case
that the qualitative assessment has removed the models with objectively larger errors in the key criteria.

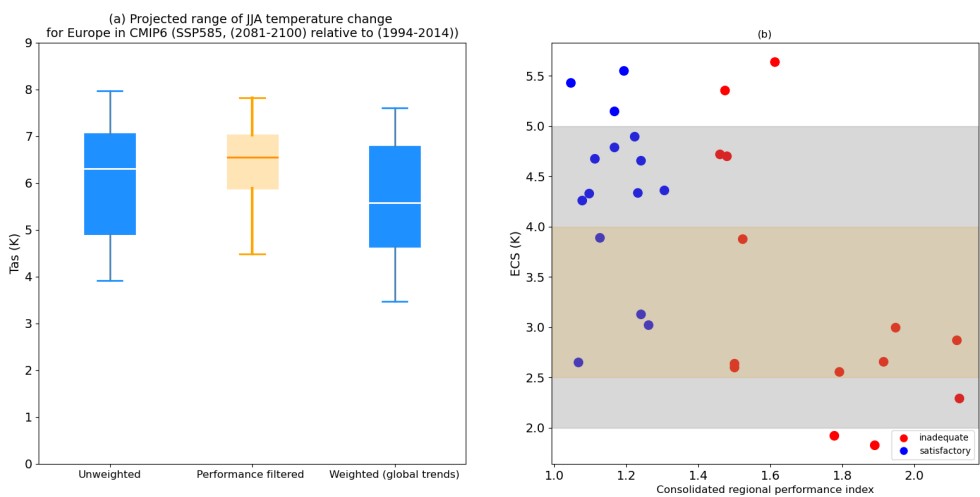

**Figure 7.** a) Projected range of JJA temperature change for Europe in CMIP6 (SSP585, (2081-2100) relative to (1994-2014)) for the raw unweighted multi-model ensemble, the performance filtered subset and the raw ensemble weighted for performance against global trends using the climWIP method. Boxes show 25th to 75th percentile. Whiskers show the 5th and 95th percentile. b) CMIP6 model ECS compared to consolidated regional performance index. The yellow bounds show the IPCC AR6 likely range for ECS, the grey bounds show the very likely range.

Figure 7a shows the difference in the range of projected temperature in the large-scale and regional performance filtered sub-set compared to the raw unweighted ensemble for JJA. The shift in the projected range for DJF in comparison is small (Fig. S2). The lower part of the range is substantially reduced for the process performance filtered in JJA (Fig 7a).

This emergent relationship between the retained models containing more of those with higher sensitivity contrasts with the existing literature of observational constraints on regional climate projections in CMIP6. There is an existing literature that has used ability of CMIP6 to capture either regional temperature trends (e.g. Ribes et al., 2022) or global trends (Liang et al., 2020; Ribes et al., 2021; Tokarska et al., 2020; Brunner et al., 2020b) that down-weight models with larger climate sensitivities, in favour of models with more modest climate sensitivities. We illustrate the contrast between this existing literature and our results, by using the methodology of Brunner et al. (2020b) to illustrate the typical constraint on projections from this literature. We use the method of Brunner et al. (2020b) (see section 3.3), applied for the global temperature trend to calculate performance weights for each model using the first ensemble member. These weightings shift the projected temperature range downwards compared to the unweighted raw ensemble (Fig. 7 a). Our emergent relationship between less robust regional projections and lower sensitivity models was unexpected and represents an apparent tension with the existing observational constraint literature based on temperature trends.

A regional consolidated performance index was created by giving the 'Satisfactory' (white), 'Unsatisfactory' (orange) and 'Inadequate' (red) flags a numerical score of 1 for 'Satisfactory', 2 for 'Unsatisfactory' and 3 for 'Inadequate'. The overall score for each model was then averaged by the total number of assessed criteria, to give an indication of how the model performed overall. Many of the models that performed well for the process based criteria do not fall within the IPCC AR6 likely range for equilibrium climate sensitivity (ECS) (Forster et al., 2021) (Fig. 7b).

Our result does not include any consideration of climate sensitivity and while these models are identified here as performing relatively well in a process-based assessment, the sub-set temperature range shown in Fig. 7 should not be viewed as a constraint that gives a more accurate projected range for Europe. Here we only highlight that more of the models that perform well in terms of regional physical processes have a higher climate sensitivity. It may be appropriate to select only the better performing the models from within the very likely IPCC range for ECS, or to retain just one of the models above this range to account for a higher impact scenario. It may also be appropriate to select models that are 'marginal' from the lower part of the IPCC very likely range. Alternative using an approach that considers regional impacts using Global Warming Levels could be applied to the sub-set, this is discussed further in section 5.

## 4.3 Sub-selection for performance and model diversity

In this section we consider how a sub-selection of a small number of example models that represent the broader characteristics of the wider filtered projection spread could be carried out. In this example application we look for a sub-set of GCMs that are both in our filtered sub-set and sample this spread. The motivating criteria is to identify models that perform well across the whole European domain and retain as much of the spread of future projections as possible. Such an approach might be adopted by those looking for a smaller subset to drive downstream models. For example, as a selection tool for a potential Regional Climate Model (RCM) matrix, as data for a pan-European assessment of food security, or any other impact needing pan-European physically coherent climate projections, where the GCMs then would provide the climate driving data.

The models from the process performance sub-set are placed into clusters of models that had clear dependencies (table 1). The Euclidean distance of the models is determined using the ClimWIP method (see section 3.3), for the comparison of model independence (Brunner et al., 2020b). Fig.S3 shows the independence matrix for the different models, which was used to create clusters of models that had dependencies, models with a Euclidean distance of =<0.6 were combined into clusters. Three models were found to not have a sufficient dependency to the other models to be placed in any cluster (see table 1). In most cases many of the models with similarities are from the same institution or are known to share significant code components, such as the same atmosphere model in the HadGEM-GC-3.1 models and ACCESS-CM2.

In this application, to maintain model diversity as far as possible, one model was selected from each of the clusters (and two from models that fell into no cluster). Using Fig. S4 to determine where the models are situated in the projected temperature and precipitation range for each region, these individual models are also selected to include as much of the temperature and precipitation range of the filtered multi-model ensemble as possible. The selection chosen for this example is illustrative and it may be appropriate to sub-select differently depending on the application of the sub-selection. The selected models for this example are shown in blue (Fig. 8).

In this section we have shown one example of sub-selection of a smaller sub-set, using the filtered models from the previous section. There are a number of different smaller sub-sets that could be selected using the information from the assessment tables (Figs. 5). Depending on the application of the sub-selection a different approach, for example, one that includes plausible outliers (e.g., models that do not have red flags in large scale criteria), may be more appropriate in order to sample high impact, low probability regional responses.

**Table 1.** Table showing models clustered based on Euclidean distance. * models were not found to have sufficient dependencies to be placed in a cluster. Selected models are shown in bold.

| No cluster* | cluster 1 | cluster 2 | cluster 3 | cluster 4 | cluster 5 |
|---|---|---|---|---|---|
| **BCC-CSM2-MR** | **GFDL-ESM4** | **EC-Earth3** | CESM2 | CNRM-CM6-1 | ACCESS-CM2 |
| **MRI-ESM2-0** | GFDL-CM4 | EC-Earth-Veg | **CESM2-WACCM** | CNRM-ESM2-1 | HadGEM-GC31-LL |
| MPI-ESM2-HR | | | TaiESM1 | **CNRM-CM6-1-HR** | **HadGEM-GC31-MM** |

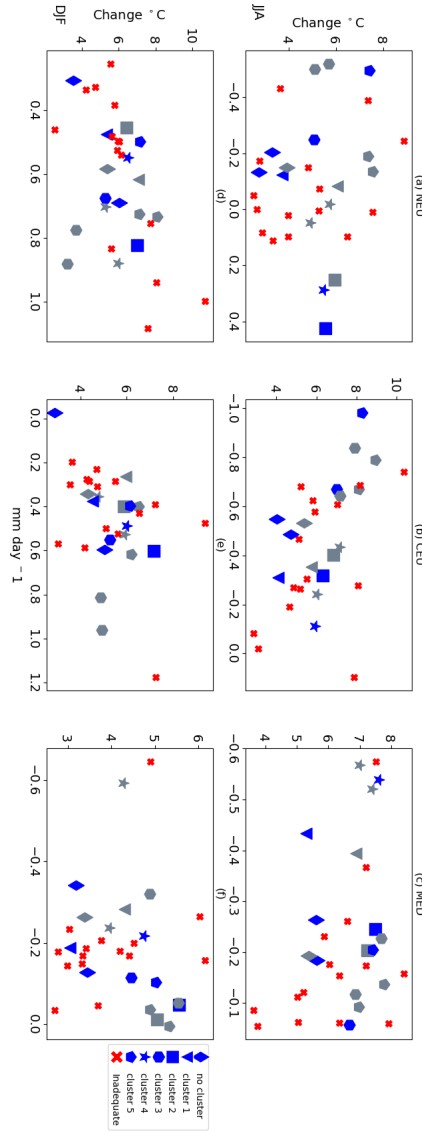

**Figure 8.** Temperature and precipitation projection range (SSP585, 2018-2100 relative to 1995-2014) for CMIP6 multi-model ensemble. Excluded models are shown as red. Models selected from each of the 7 clusters in table 1 shown as blue. Models from the process performance filtered subset not selected shown in grey Models from the same cluster are indicated by symbol.

## 5 Discussion

An overall aim of this study is to provide an assessment of CMIP6 models that can be applied by users that wish to create filtered sub-sets for Europe, for a range of applications and also wish to remove the least representative models. The assessment information could be applied to a filtering approach that is tailored depending on the criteria of interest. The assessment used in this study combines a qualitative and quantitative approach. To some extent there is always a degree of subjectivity when grading models for performance, even where more objective techniques are used, such as clustering based on evaluation statistics, (e.g. seasonal RMSE, correlation, bias as used for the blocking frequency here) there is still the difficulty of identifying where the thresholds are for what is a 'Satisfactory' or 'Inadequate' model should lie, and the assessment of the relative importance of one metric versus another (Knutti, 2010). The assessment of a 'Satisfactory' model will also inevitably be relative to the performance of other models in the ensemble. In the approach shown here, where the quantitative thresholds were used to guide the model classifications, these thresholds were largely determined from the distribution of performance for the ensemble. It has also been an aim throughout to maintain consistency in the way that the classifications are applied for each of the assessment criteria. In practice this can be difficult to achieve as many of the GCMs generally capture some of the large-scale processes relatively poorly (e.g. blocking frequency and CEU precipitation) in comparison to others for each criteria, and also due to the difficulty in evaluating others (e.g. AMOC).

A further challenge is that not all models have been assessed against all criteria. Analyses that assessed storm tracks, blocking frequency and the AMOC provide valuable further information regarding the performance of the models, but were not available for every model in the study, therefore it was necessary to consider whether a model should be eliminated on the basis of one of these criteria, when other models where their performance was unknown may be kept in the selection. It was found to be the case that the flags for exclusion did not occur in isolation, severe errors (red flags) for large scale circulation, storm tracks and blocking frequency often occurred in more than one criterion (or in some cases alongside multiple orange flags). Severe errors (or 'Inadequate', i.e., those flagged red) in the AMOC, another criterion where data was limited, were due to a very weak representation of this feature and where this is the case a severe cold bias in the SPG region was also present (NESM3).

Considering the regional impact of eliminating the models flagged as 'Inadequate' (flagged red, Fig. 5); the lowest temperature response models are excluded in summer (JJA) for the NEU, CEU and MED (Fig. 8 ). For summer rainfall, in the NEU and MED, many of the models showing a more neutral change in rainfall are excluded. Greater warming is generally linked to stronger summer drying and increased winter precipitation. The exclusion of many models with a more modest projected temperature increase also excludes many of the more neutral projected changes in precipitation.

Filtering from the CMIP6 ensemble by excluding the least realistic models for Europe leads to the removal models throughout the projected temperature range but removes more of the models that have a more modest response. The retention of higher

sensitivity models is due to more of the higher sensitivity models demonstrating a greater skill for reproducing regional processes. The revised temperature projections for the filtered GCMs for each region leads to a shift upwards in the median of the projected JJA temperature range, due to more of the higher sensitivity models performing well against the process-based criteria (Fig. 7). This may represent a particular challenge for potential applications where sampling regional climate responses in the lower end of the IPCC climate sensitivity (ECS) range is required, as many of the CMIP6 models in the lower part of the ECS likely range were excluded by our processed based assessment (Fig. 7b). Using the IPCC AR6 likely range for ECS (and or TCR, Hausfather et al. (2022) has also been suggested as an approach to model screening for the CMIP6 ensemble. Other regional sub-selection studies for CMIP6 have eliminated models with high global sensitivity (Mahony et al., 2022). Any assessment that excludes models based on both the performance criteria here and metrics like global climate sensitivity or global trend criteria (which tends to exclude models with higher climate sensitivities) will be left with only a small sub-set of "adequate" models. This apparent tension is likely to be less evident where Global Warming Levels are instead adopted. Using this approach, when adopting a selection based on climate performance, such as presented here, would enable a broader set of "adequate" realisations to be explored.

Our results contrast with the existing literature based on evaluation against historical temperature trends (Liang et al., 2020; Ribes et al., 2021; Tokarska et al., 2020). Many of the models that score well against process-based criteria, have a higher ECS. ECS is not considered in this study as a sub-selection criterion, because the focus of this work was on the assessment of how well models capture the main regional climate processes. Links between plausibility of CMIP6 projections, based either on their historical global or regional temperature trends or climate sensitivity (Hausfather et al., 2022) are well established in the literature for CMIP6 (Ribes et al., 2021; Liang et al., 2020; Tokarska et al., 2020). When the raw ensemble is weighted against performance for global trends (Fig.7a) the effect is to shift the temperature range downwards. This shift for our raw ensemble is not as large as typically seen for in other studies for global trends (e.g. Liang et al., 2020; Ribes et al., 2021; Tokarska et al., 2020), this may be due to our using a single ensemble member for this study, some differences in methodology, or due to summer warming in Europe being thought to be about 30% higher than the annual mean global warming (Ribes et al., 2022).

Ribes et al. (2022) find a constrained regional projection range for mainland France for ssp585 (5.2 to 8.2 °), that is similar to the projected range for summer of our performance filtered subset, using a combination of modelling results and observations. Our upper 95th percentile and lower 5th percentile is a little lower than this (for a pan-European range), our median for the performance filtered range is very similar to their central estimate of 6.7 °C (Fig. 7a). For assessments of model performance against historical temperature trends, where the regional trends are also taken into account there may be less of a tension with our assessment, than is the case with those that are based on global trends alone (e.g. Liang et al. (2020); Ribes et al. (2021)).

## 6 Conclusions

We provide an assessment of regional processes and biases (Fig. 5) for a multi-model ensemble for CMIP6 that can be used to inform sub-selection for the European region. This can be used to aid the creation of bespoke sub-selections for a particular application (e.g., sub-selection of a small number of representative ensemble members, for downscaling or impact assessments), alternatively the sub-sets that have been demonstrated here can be also be used directly.

Filtering an ensemble of CMIP6 models based on performance against key processed based criteria results in the projected temperature range being shifted upwards. This is due to the removal of a larger proportion of the lower climate sensitivity models, that do not perform adequately against the assessment criteria. We also find that many of the higher sensitivity models score well against the process-based assessment and that these models are better able to represent the features of the European climate. It is not clear whether the emergent relationships we found (between better models and higher sensitivities) is circumstantial or reflects an underlying physical basis. If it reflects an underlying physical relationship (where atmospheric processes needed to capture regional feedbacks also drive stronger climate feedbacks) then this might imply greater confidence in higher end regional changes. If on the other hand the sampling of higher sensitivity models is circumstantial (simply due to chance), this represents a challenge as there are few CMIP6 models that sample the central and lower end of the IPCC AR6 likely climate sensitivity range. This remains an open question, which we have not been able to resolve in this work.

Our results highlight a tension for regional sub-selection between performance against the global temperature trend and the ability of the models to capture the features of the regional climate in the CMIP6 multi-model ensemble. For cases where changes in temperature are not the only variable of interest (or the primary concern) many of the higher sensitivity models are likely to provide more reliable information regarding the future climate. Potential users of regional climate projections should be aware that there is a potential tension between constraints from large scale temperature change/climate sensitivity and the assessment of regional processes, for Europe at least.

*Code and data availability.* The code used to apply the climWIP method is publicly available via the ESMValTool (https://docs.esmvaltool.org/en/latest/recipes/recipe_climwip.html)

The data used in this study is available through the ESGF data portal at https://esgf-node.llnl.gov/projects/esgf-llnl/. Details of the models are available from https://wcrp-cmip.github.io/CMIP6_CVs/docs/CMIP6_source_id.html

Further assessment plots for the models used in this paper are available on github https://github.com/tepmo42/cmip6_european_assessment as well as spreadsheet of all available assessments (for Europe) carried out for CMIP6 models to date.

## Appendix A: Appendix

### A1 Annual precipitation cycle

The annual precipitation cycle was assessed as a regional criteria for each of the European land regions (Gutiérrez et al., 2021) (see Fig.S1). The precipitation cycle for each model was assessed against the EOBs data as monthly means (see figure A1) using the baseline period of 1995-2014. A combination of the correlation and RMSE for each of the 4 seasons is used to assess whether models should be categorised either as satisfactory or unsatisfactory.

In order to sort the models into categories the seasonal RMSE and correlation were used as a guide (Fig. A1 b-e), it was observed that in most regions a poor correlation with the observed cycle had a large seasonal RMSE compared to other models (>0.75mm day$^{-1}$), in at least one season. The exception was in CEU where most models had a poor correlation with the observations. The models with a relatively low error of less than 0.6mm day$^{-1}$ for all four seasons were classified as satisfactory. Models with a RMSE of greater than 0.75mm day$^{-1}$ in one season were classified as unsatisfactory.

In CEU the models with a substantially negative correlation (approx -0.25) had a large seasonal RSME (>1mm day$^{-1}$) and very poor agreement with the seasonal cycle where they essentially showed a strong seasonal drying in the wet season for CEU (figure A1). These models are classified as excluded.

There were some models (RMSE >0.6mm day$^{-1}$ <0.74mm day$^{-1}$) that did not fall immediately any classification, Where the RMSE error in all seasons was < <0.74mm day$^{-1}$, the model performance was generally satisfactory. However, in the case of some models closer this threshold with some were 'Unsatisfactory', if they had a lower correlation with the EOBS data.

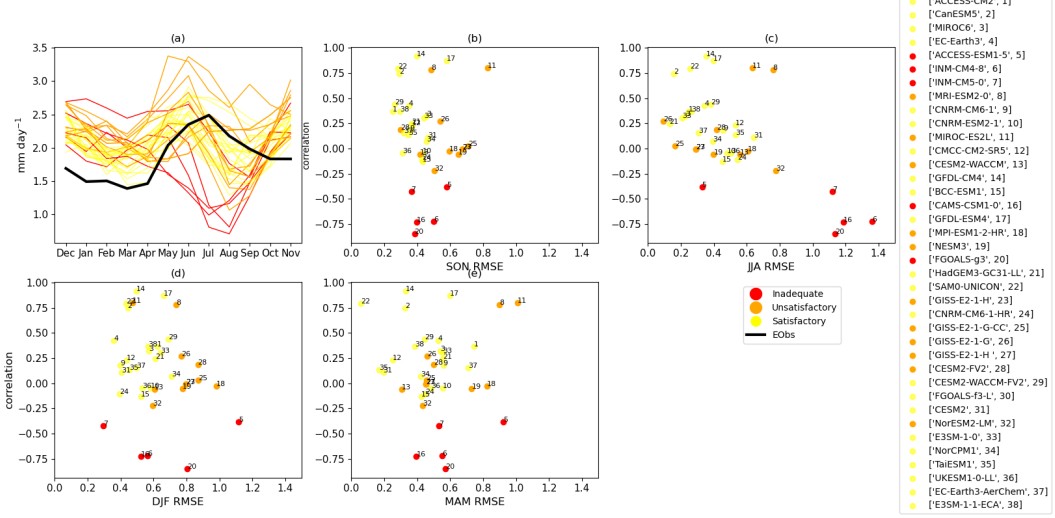

**Figure A1.** Precipitation annual cycle for CEU (top left), model comparison with EOBS (shown as solid black line). Correlation (over 12 months) and seasonal RMSE for each model. Monthly averages are taken over a 20-year climatology (1995-2014). The RMSE and correlation are calculated from the monthly averages

## A2    Sea Surface Temperature bias

Seasonal average Sea Surface Temperatures (SSTs) were assessed for each of the models using the HadISST1 reanalysis (Fig A2) for the baseline period 1995-2014. Surface skin temperatures from the atmospheric models were used, the corresponding ice concentration fields from the atmosphere model were only available for a smaller number of models. To estimate the ice extent and avoid errors in the assessment of the SST bias in areas affected by ice, a seasonal average ts < 0 is used as a proxy for the 5% ice concentration to mask these areas. The areas masked by this proxy is compared to the extent of the 5% ice concentration in the models and found to be a good approximation. As the area affected by ice is approximated this is not compared directly to the 5% ice field from HadISST1 for the assessment, however where the mask areas are significantly larger than the 5% ice concentration in HadISST1 (Fig A2, bottom right), a large cold bias in these areas is inferred (figure A2). This bias in sea ice and the SST surrounding northern Europe is found to be well captured by the large-scale near surface temperature bias (see section A3). Therefore it is noted here, as an important consideration for the European climate, but not included explicitly in the assessment of the NA SST error classifications. For the NA SST assessment we focus on errors in key regions of the NA for the European climate.

The NA SST assessment is based on two key areas of the NA, the subpolar gyre (SPG) and Gulf Stream northwest corner (GS) regions. These have been selected from Ossó et al. (2020), who identified a North West region of the North Atlantic GS as important for weather patterns over Europe, and Borchert et al. (2021b) to define the SPG region, which has previously been shown to modulate the probability of occurrence for summer temperature extremes in central Europe (Borchert et al. (2019) ; Fig. S7). These regions as well as their gradient has been demonstrated to carry relevance for dynamical atmospheric influences of NA SST on European summer climate (Carvalho-Oliveira et al., 2022)), highlighting their relevance in the context of this study. During a qualitative inspection of the models (see A2) these regions were also identified as often areas to routinely show a substantial bias in the models.

A small number of models had extensive areas with a very large winter negative SST bias (Fig A2, bottom row), this results in a substantial over estimation of winter ice extent to the north of Scandinavia and around Greenland. NESM3 and CAMS-CSM1-0 have a large widespread negative bias that extends beyond the the regions of sea ice to the NA and SPG. (Fig A2). The models with the largest SPG RMSE are NESM3, CanESM5, CAMS-CSM1-0 (shown in Fig 1) and FGOALS-f3-L. In addition, FGOALS-f3-L has an RMSE for the GS region of more than twice the ensemble mean RMSE. These models are all flagged as 'inadequate'.

A number of models also had areas with substantial but limited areas of warm bias ( >6K ) in the area around the Gulf Stream and larger areas in the SPG ( >3K) e.g. CESM2, INM-CM50, NorESM2-LM (Fig A2). In addition, these models also have areas cold bias in the SPG, this combination of warm and cold biases in different areas also results in a poor representation of the SPG temperature gradient. These models also had an RMSE larger than the ensemble mean in the SPG assessment

region and were classified as 'unsatisfactory'. The INM models are an exception in the 'unsatisfactory' category in terms of not having a large SPG error, however these two models have some of the largest errors in the GS region, with the exception of FGOALS-f3-L.

Satisfactory models had a lower bias in all areas, some had small areas with a larger bias (often around the Gulf Stream or in some parts of the SPG, (e.g. ACCESS-CM2), but the effect of these areas did not prevent a reasonable representation of the SPG gradient. The models classified as 'satisfactory' all have a RMSE in the assessed region that is less than the ensemble mean.

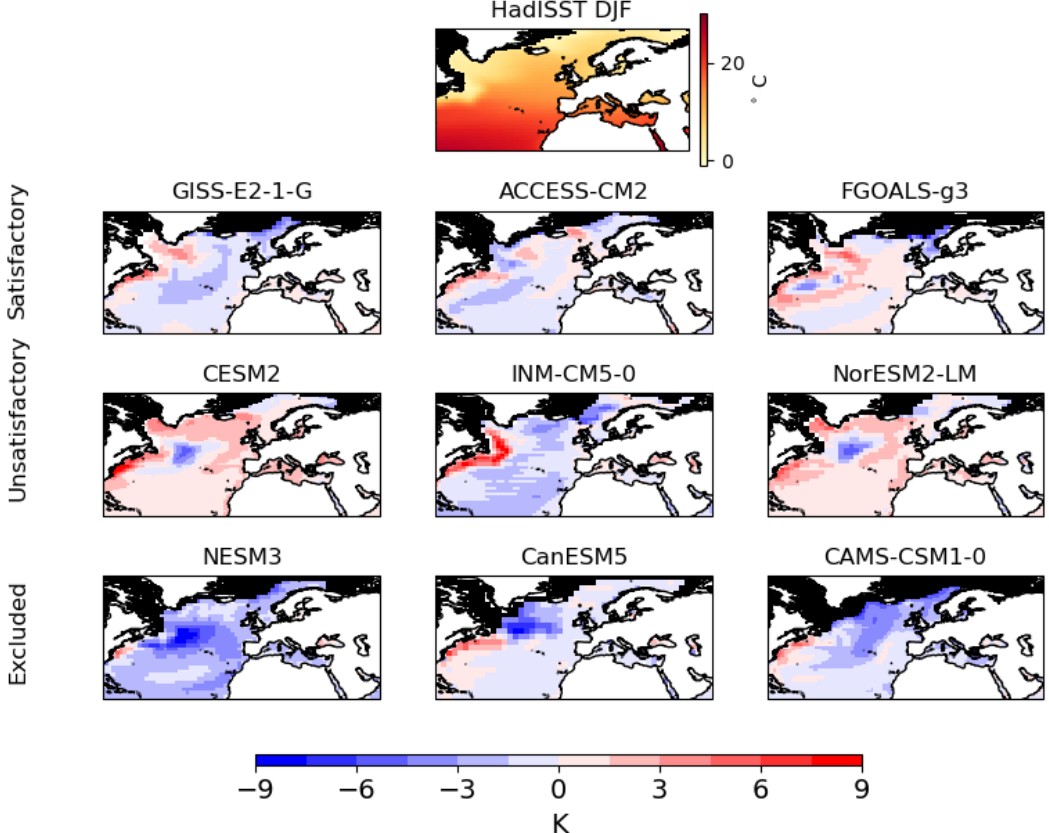

**Figure A2.** Model SST bias (compared to HadISST) for DJF. Seasonal average calculated for a 20 year climatology (1995-2014). Areas where the model SST < 0 are mask in black (this was found to approximate to 5% ice concentration). Top row shows the HadISST and 5% ice concentration field.

In JJA the satisfactory models again had smaller areas of bias (>3K) around coastal regions, but these were not widespread (Fig. A3). Models flagged as satisfactory also had SST RMSE in the SPG and GS regions that were less than the ensemble mean error. Although the model errors in the SPG ans GS are satisfactory there is a cold bias in the SSTs in the Norwegian and Barents Sea in the FGOALS-g3 (and GISS-E2-1-G) model. In the FGOALS-g3 model there is also an excess in the sea ice extent in the Barents region. While the SST assessment has been focused on the NA SST region it is noted that biases in this region are also important for the European climate. These biases are captured in the model classification for temperature bias (which includes near surface temperature bias over the ocean).

The unsatisfactory models had larger regions with a substantial cold bias in the SPG and/or larger biases in the GS region, that were larger than the ensemble mean. The CAMS-CSM1-0, CanESM5 and FGOALS-f3-L models with the largest SPG errors were classed as 'inadequate'.

## A3    Near surface temperature bias

### A3.1    large scale bias

The model near surface temperatures is compared to ERA5 reanalysis (Fig A4) for the baseline period 1995-2014. These were assessed for the large-scale domain (including surrounding areas over the NA, Norwegian Sea, Barents Sea and nearby Arctic regions) criteria (Fig.A4,) and also more specifically for the land points of each SREX region (see in the following section A3.2).

For the large scale assessment there is inevitably some overlap with the assessment of SST temperatures as near surface temperature over North Atlantic regions is taken into account. The large scale qualitative assessment considers whether there is widespread areas of temperature bias in land regions of Europe or in other regions where they could be expected to have downstream impacts, e.g. nearby land areas NA, or other ocean regions nearby the European land areas. A more widespread bias as opposed to a smaller more regionally based temperature bias, indicates an issue with the large-scale processes that will affect all the European regions, while a more local area of bias is likely to indicate issues related to processes in a particular region. Where biases in land regions are found in more than one European region however these are likely to also indicate an issue that may affect the whole European area. The RMSE error over the region as a whole and each of the land regions is taken into consideration for the model calculation alone with a more qualitative assessment of regions of bias.

For JJA, MIROC6 has a large widespread positive summer bias over European land regions, north Africa and Greenland, this bias is largest in the CEU and MED, but this is not extended over the NA where there is a cool bias. The warm bias in the MED and CEU regions is exceptionally large (>8K in some areas), but it is not limited to these regions, with a smaller but still substantial bias for all land regions (Fig A4). The RMSE is the largest in the model ensemble for the whole region and the northern and central regions. MIROC-ES2L has a similar pattern of errors as MIROC6 (although not quite as large, still more

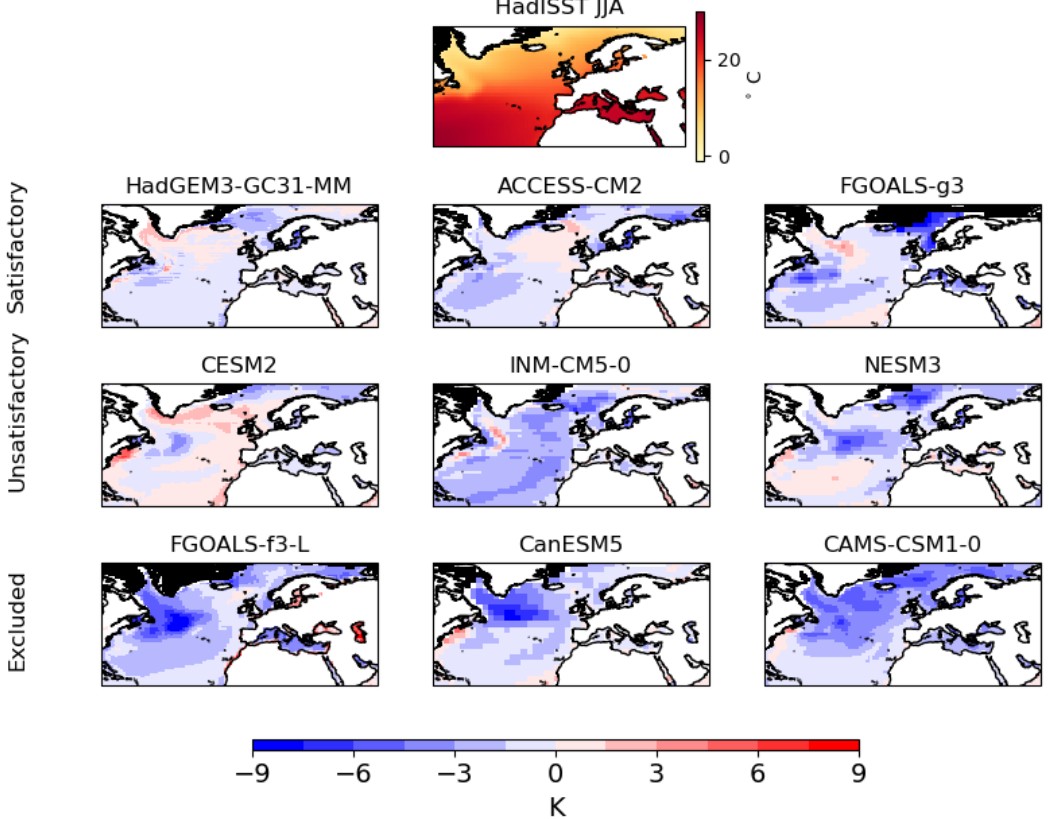

**Figure A3.** Model SST bias (compared to HadISST) for JJA. Seasonal average calculated for a 20-year climatology (1995-2014). Areas where the model SST < 0 are mask in black (this was found to approximate to 5% ice concentration). Top row shows the HadISST and 5% ice concentration field.

the 1.5 times the ensemble mean RMSE. CAMS-CSM1-0 has a large widespread bias negative bias in all areas of Europe, this model also has a large cold bias in JJA for both for land and SST. It has one of the largest RMSE for the large-scale region and the largest in the ensemble for northern Europe. FGOALS-g3 also had widespread biases with an unusual pattern showing an area of exceptionally large cold bias to the north of Scandinavia and the UK (>8K), while also having a substantial warm bias in the eastern area of CEU (4-6K around the black sea area). The RMSE for the whole region is above average, but not exceptionally large compared to the rest of the ensemble, this is largely due to a relatively small bias in the NA, as noted in the SST assessment. The RMSE error in the central European region is more than 1.5 times the ensemble mean. The additional area of large low bias in the Norwegian and Barents Sea area, with the resulting excessive sea ice (see Fig.A3), has led to this

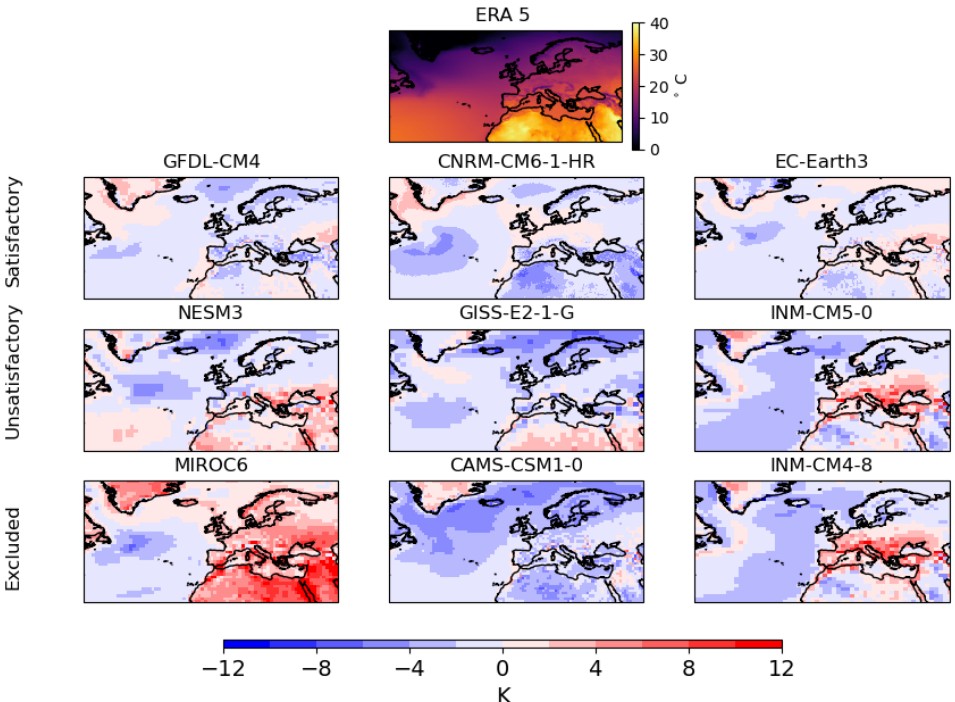

**Figure A4.** Model temperature bias for the large-scale domain. Seasonal JJA average calculated for a 20 year climatology (1995-2014).

model also being rated as 'inadequate'. The INM-CM4-8 model has a large positive bias in both the central and Mediterranean regions, and RMSE for both these regions and the SPG is more than 1.5 times the ensemble mean error; therefore this model has also been classified as 'inadequate'.

Examples of models classified as 'Unsatisfactory' for JJA bias, include NESM3, GISS-E2-1-G and INM-CM4-8 (Fig A4).
NESM3 has a substantial warm bias in eastern CEU and MED regions (> 4 in some areas) and areas of cold bias in the NA (4-7K). GISS-E2-1-G has substantial more widespread areas of cold bias (Fig. A4). The INM-CM5-0 model has a substantial warm bias in the central European region and SPG area. Its overall RMSE for the large-scale area is large than the ensemble mean RSME, this model is classified as 'unsatisfactory'.

Examples of 'satisfactory' models with a bias of </= 2K in most regions for JJA and limited regions with bias of up to  4K in limited areas, include GFDL-CM4, CNRM-CM6-1-HR and EC-Earth3 ( Fig. A4 (top row)) Models classified as 'satisfactory' had a large scale RMSE that was less than or close too (slightly above) the ensemble mean RMSE.

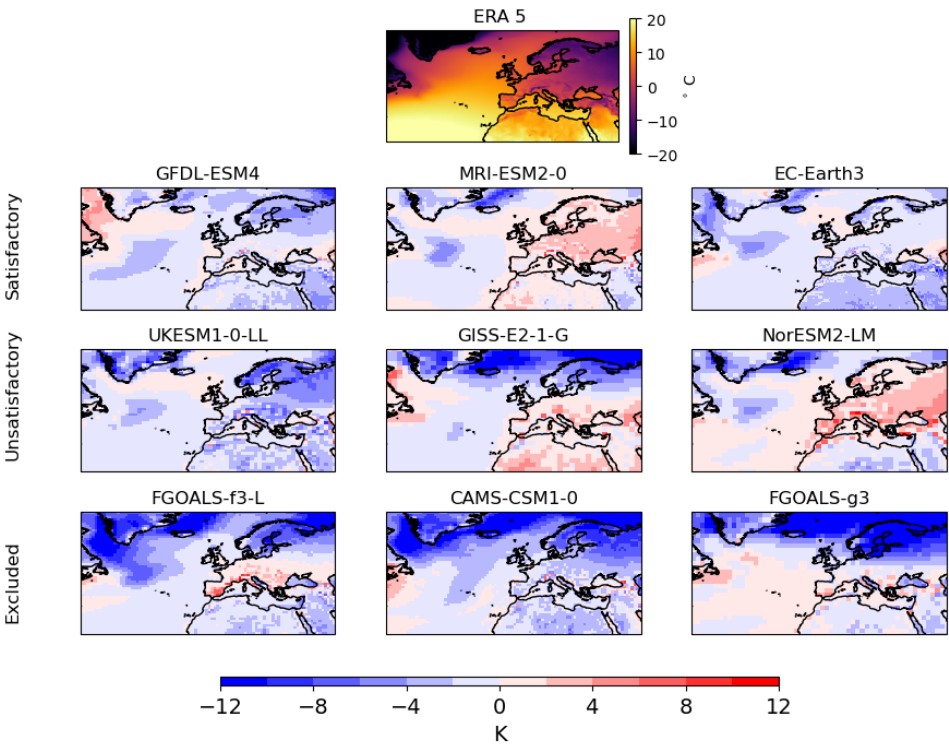

**Figure A5.** Model large-scale temperature bias for the large-scale domain. Seasonal DJF average calculated for a 20 year climatology (1995-2014).

For DJF the cold bias in the models that are classified as inadequate is pronounced, especially in northern European areas (Fig. A5). These models all had an RMSE for the large-scale area that was more than 1.5 times the ensemble mean RMSE. In the case of FGOALS-g3 it was more than twice the ensemble mean error.

The unsatisfactory models included those with substantial cold bias in areas than while not directly over European land regions can be expected to have some downstream impacts on them (e.g., NESM3, GISS-E2-1-G). In several cases substantial biases are present in the land regions of interest (e.g., NorESM2-LM). The models classified as 'unsatisfactory' all had RMSE errors large than the multi-model mean. The only exception is the UKESM1-0-LL, which had RMSE for the large-scale area that was slightly lower, but substantial errors in two European land regions (northern and central Europe) that were among the largest in the multi-model ensemble. 'Satisfactory' models had smaller biases over all regions and a RMSE for the large-scale that was smaller than the multi-model ensemble mean.

### A3.2 European land regions

In addition to the large scale assessment the three IPCC AR6 land regions (Gutiérrez et al., 2021) were individually assessed to identify land areas of seasonal temperature bias. The spatial mean seasonal RMSE for all land points in each region was calculated and used as a guide for assessment, along with a visual inspection of the spatial temperature bias. A small number of models were classified as 'Inadequate' for individual regions due to areas with a large local bias, that were not excluded due to a temperature bias in the large-scale assessment. These models may be considered as 'Inadequate' only for the region.

The RMSE for each region is used to classify the models, for JJA the thresholds were for 'satisfactory' < 2.5K, 'unsatisfactory' >2.5K but < 4K, 'inadequate' >4K. As is the case in determining any threshold there is a degree of subjectivity and these thresholds are based on the relative performance of the models across the ensemble. For DJF the thresholds were the same except that the threshold for 'inadequate was increased to >5K.

### A4 Atlantic Meridional Overturning Circulation

The representation of the AMOC is still considered to be deficient even in state of the art GCMs, where its associated climate impacts are also thought to have been underestimated (Zhang et al., 2019). In addition due to the limited availability of observational data there is still considerable uncertainty in the recent AMOC evolution (Menary et al., 2020), and accurate assessment of the AMOC in climate models remains challenging. For this study some assessment of the AMOC is considered to be important due to its potential role in future changes in the European climate. The aim is to identify and flag the poorest models with large errors in the representation of the AMOC compared to the observational data from the rapid array. Examples of the overturning stream function for each model shown (Fig A6) is calculated using the method of (Menary et al., 2020).

NESM3 and IPSL-CM6A-LR both show poor agreement with the observational data, with a consistently weak AMOC (Fig A6). NESM3 was classified as 'Inadequate', the impact of the AMOC on the NA SSTs is also flagged due to a large cold bias. The AMOC for IPSL-CM6A-LR is flagged as 'Unsatisfactory', the error may impact on the representation of the NA, but the impact on the reliability of future projections is not clear, a similar error was present in CAMS-CSM1-0, which is also flagged as 'Unsatisfactory'. In contrast the NorESM2-LM model has a consistently strong AMOC through the historical period, with a rapid decrease in more recent years which is not seen in the observational data, this model is also classified as 'Unsatisfactory'. The other models for which AMOC data was available are classified as satisfactory (e.g.INM-CM4-8 and INM-CM5-0), as they do not show a large deviation from the observations.

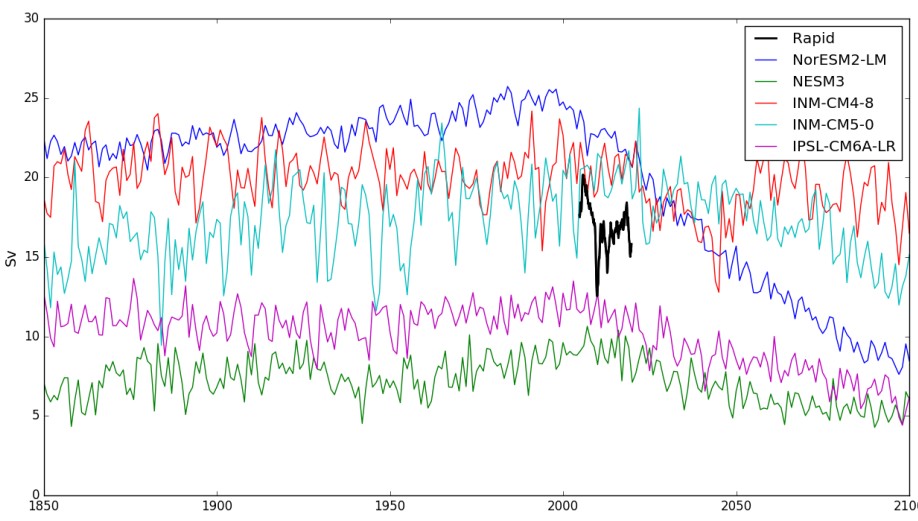

**Figure A6.** AMOC strength at 1000m (from v - velocities compared to rapid array (annual mean Sv) at 26°N, AMOC data from Menary et al. (2020)

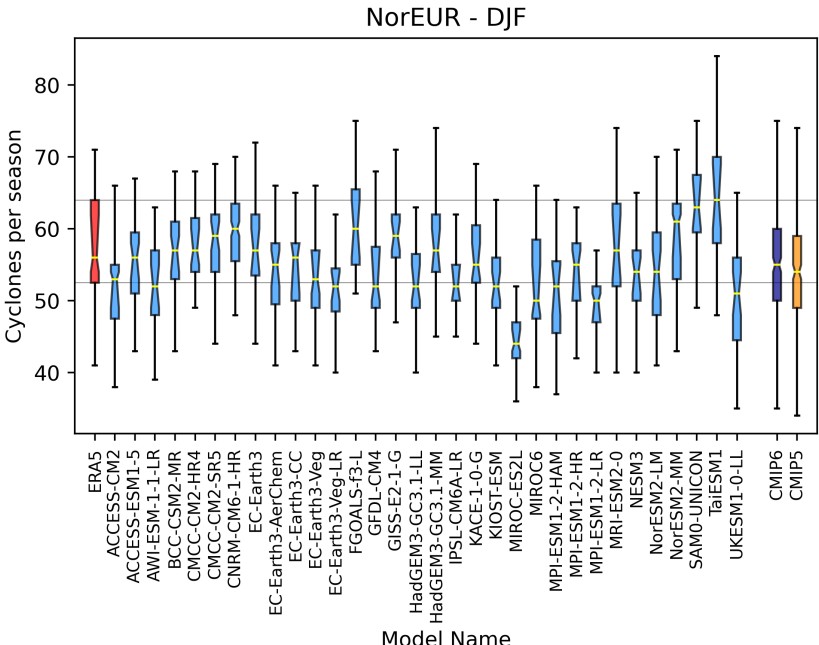

**Figure A7.** 'Boxplots of cyclone numbers per DJF season for NEU region (co-ordinates). Boxes are shown for ERA5 (red), 32 CMIP6 models (light blue), and the CMIP6 (purple) and CMIP5 model ensembles (orange). Boxes extend to the 25th and 75th percentile of the distributions, with whiskers extending to 1.5 times the inter-quartile range. Horizonal yellow lines indicate the medians. Notches around the medians show its uncertainty based on 10,000 random resamples. Horizontal gray lines indicate the ERA5 25th and 75th percentiles

## A5 Storm Tracks

### A5.1 Regional assessment

The storm tracks were also assessed regionally to determine whether the number and variability of the cyclones in a particular region were captured satisfactorily by the models. This used the analysis of Priestley et al. (2020) for the individual European regions. The baseline time period used for this assessment is 1979/80-2013 for CMIP6 (1979/80-2004/05 for CMIP5) and the model data is compared to ERA5.

Where the 25th and 75th percentile of the range overlapped and was similar in size to the ERA5 data the model was classified as satisfactory. If the interquartile range of the model had no overlap with ERA5 data or the size of the interquartile range was substantially smaller than the model was categorised as unsatisfactory for the region (see Fig.A7. Models are not excluded based on the regional analysis, these flags are for information only.

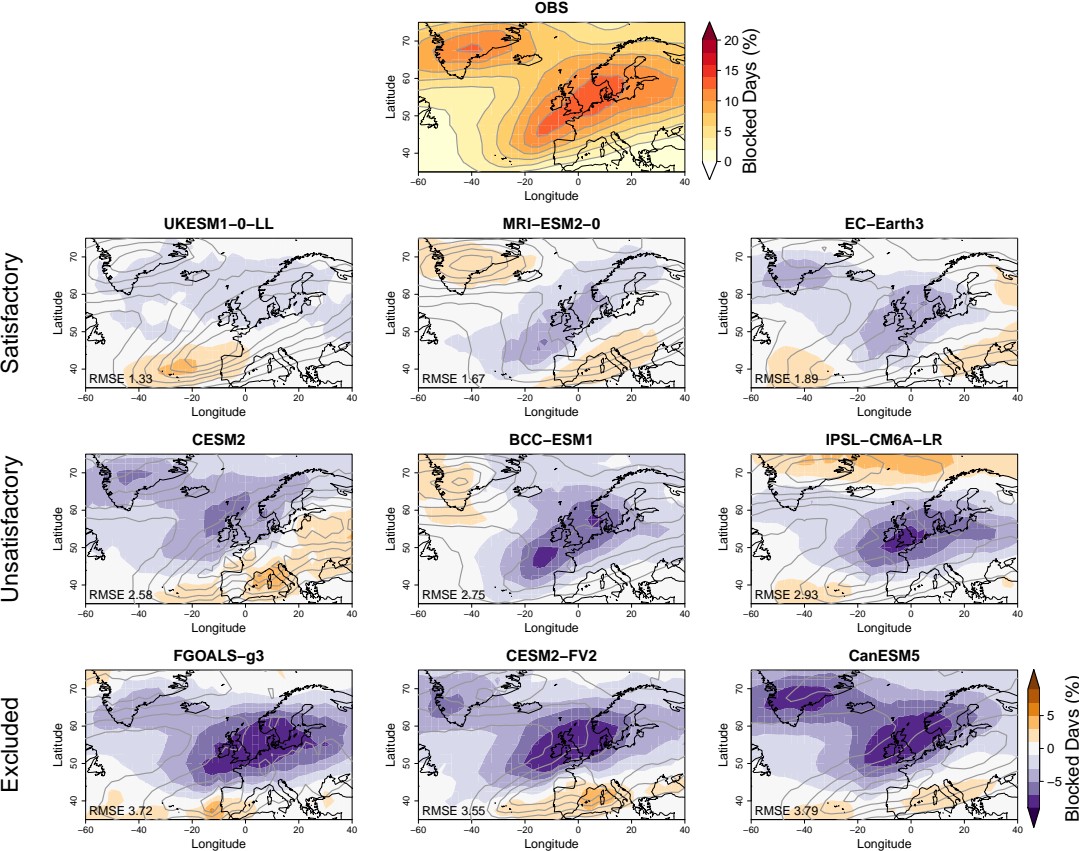

**Figure A8.** Examples of DJF Blocking Frequency classifications for a sample of individual models..

## A6 Blocking frequency

Atmospheric blocking is a recurrent weather pattern typically occurring in the mid-latitudes at the exit of storm track (Rex,
1950; Pelly and Hoskins, 2003). It is characterized by a high-pressure low-potential vorticity quasi-stationary large-scale
anomaly which is able to "block" or divert the movement of the traveling cyclones, creating anomalous weather in its underlying
region. One challenging issue for the climate community is the struggle that weather and climate models have in reproducing
the observed frequency of atmospheric blocking (D'Andrea, 1998; Masato et al., 2013). Indeed, state-of-the-art climate models
are known for underestimating the frequency of atmospheric blocking especially over the Euro-Atlantic sector, albeit notable
improvements have been observed with the last generation of models (Davini and d'Andrea, 2020).

In this work atmospheric blocking is identified with an objective index based on the reversal of the daily geopotential height
gradient measured at 500hPa, making use of the blocking index developed by Davini et al. (2012)). The index is the 2-d
extension from 30°N to 75°N of the canonical definition by Tibaldi and Molteni (1990). However, we here adopt a blocking

definition which includes a third supplementary condition south of the blocked region aimed at excluding the low latitude
blocking events (see Davini et al. (2012) for details). Defining Z500 as the daily geopotential height at 500hPa interpolated on
a common regular 2.5°x2.5°grid, three meridional gradients are considered:

$$GHGS(\lambda_0, \phi_0) = \frac{Z500(\lambda_0, \phi_0) - Z500(\lambda_0, \phi_S)}{\phi_0 - \phi_S},$$ (A1)

$$GHGN(\lambda_0, \phi_0) = \frac{Z500(\lambda_0, \phi_N) - Z500(\lambda_0, \phi_0)}{\phi_N - \phi_0}$$ (A2)

$$GHGS2(\lambda_0, \phi_0) = \frac{Z500(\lambda_0, \phi_S) - Z500(\lambda_0, \phi_{S2})}{\phi_S - \phi_{S2}}$$ (A3)

and $\phi_0$ ranges from 30°N to 75°N while $\lambda_0$ ranges from 0° to 360°. $\phi_S$= $\phi_0$ - 15°, $\phi_N$= $\phi_0$ + 15°, $\phi_{S2}$= $\phi_0$ - 30°. Instantaneous
Blocking is thus identified when:

$$GHGS(\lambda_0, \phi_0) > 0 \qquad GHGN(\lambda_0, \phi_0) < -10 \, \text{m/°lat} \qquad GHGS2(\lambda_0, \phi_0) < -5 \, \text{m/°lat}$$ (A4)

As done by Davini and d'Andrea (2020), no spatial or temporal filtering is applied.

29 CMIP6 models are taken into consideration, considering the time window 1961-2000. In order to define an objective
method to classify into categories the atmospheric blocking bias over the Euro-Atlantic region (60°W-40°E, 35°N-75°for
winter and 60°W-40°E, 45°N-75°N for summer) two basic metrics has been introduced: the RMSE and Pearson correlation
coefficient, evaluated against ERA5 reanalysis. Both RMSE and Pearson correlation coefficients are then standardized and
used as non-dimensional parameters to perform a k-means clustering (Michelangeli et al., 1995) with k=3. In this way, climate
models showing similar bias in both magnitude and pattern are clustered together, taking into account not only the size of the
bias but also its shape. An example of the classification is provided in Figure A8.

*Author contributions.* Tamzin Palmer: conceptualization, data curation, formal analysis, Investigation, methodology, software, supervision, validation, visualization, writing - original draft, writing -review and editing.

Carol McSweeney: conceptualization, funding acquisition, methodology, project administration, supervision, validation, writing -original draft, writing - review and editing.

Ben Booth: conceptualization, funding acquisition, methodology, project administration, supervision, validation, writing -original draft, writing - review and editing.

Matthew Priestley: conceptualization, data curation, formal analysis, software, Investigation, methodology, validation, visualization, writing - original draft, writing -review and editing.

Paolo Davini: conceptualization, data curation, formal analysis, software, Investigation, methodology, validation, visualization, writing - original draft, writing -review and editing.

Lukas Brunner: conceptualization, methodology, validation, visualization, software, writing -review and editing.

Leonard Borchert: conceptualization, methodology, validation, writing -review and editing.

Matthew Menary: validation, data curation, software, writing - review and editing.

*Competing interests.* We declare that the authors have no competing interests or conflict of interests.

*Acknowledgements.* This work was carried out as part of the EUCP project which is funded by the European Commission through the Horizon 2020 Programme for Research and Innovation: Grant Agreement 776613.

We acknowledge the World Climate Research Programme, which, through its Working Group on Coupled Modelling, coordinated and promoted CMIP5 and CMIP6. We thank the climate modeling groups (particularly those listed in tables ?? and S1) for producing and making available their model output, the Earth System Grid Federation (ESGF) for archiving the data and providing access, and the multiple funding agencies who support CMIP, CMIP6 and ESGF. Terms of use and further instructions can be found at https://pcmdi.llnl.gov/.

Matthew B. Menary was supported by the European Union Horizon 2020 project 4C, Climate-Carbon Interactions in the Coming Century (grant 821003) and the ANR-Tremplin ERC project HARMONY (ANR-20-ERC9-0001). Leonard Borchert was also funded by the ANR-Tremplin ERC project HARMONY (ANR-20-ERC9-0001) .

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
