# Peer review of "Performance based sub-selection of CMIP6 models for impact assessments in Europe"

_Earth System Dynamics, 2022_

## Author Comment (AC3)

**Reviewer 2 response**

Recommendation: Major revision

The authors assessed CMIP6 models in terms of their performance and diversity in simulating several variables, e.g., temperature, precipitation, and circulation, over Europe. Based on the assessment, they created sub-sets of CMIP6 models, which can be used for downscaling or impacts assessments. The approach can also be applied to other regions of the world. The topic is important and falls within the scope of the journal. The manuscript is generally well written. My major concern includes: the assessment of CMIP6 models did not well consider the link between the model's ability to simulate historical climate and future climate change. The assessments are overly dependent on subjective assessment criteria. Detailed comments are laid out below.

We thank the reviewer for this overall positive and constructive response.

Major comments:

1. No link was established in terms of the model's ability to simulate the historical climate and the projected changes. Thus, the models that can better reproduce historical climate may not necessarily generate a more reliable projection of future climate. After excluding the least realistic models, the filtered CMIP6 models show higher sensitivity. Is the result reasonable?

It is correct that we do not attempt to explicitly link baseline performance to the credibility of future projections. What we do suggest, is that there are a number of issues around using climate model projections from models which do not behave realistically in terms of key large scale regional climate characteristics in the baseline climate. Here the question is not whether well performing (better) models can offer a (more) reliable projection, but whether those models that we know to be particularly unrealistic in terms of the key large scale climate characteristics that determine the regional weather, and its variability can offer useful information about projected future climate to the climate impacts community. An increasing body of literature does link short comings in the ability of a model to realistically represent an observed baseline to being an indicator that the models' future projections are less reliable (e.g., Whetton et al., 2007; Overland et al., 2011; Lutz et al., 2016; Ruane and McDermid, 2017; Jin, Wang and Liu, 2020; Chen et al., 2022).

Having identified models that we consider particularly unrealistic to arrive at a filtered subset, we then explore what that means for the range of future projections. We find that the better-performing filtered subset happens to contain a higher proportion of higher sensitivity models. This study is not intended to present an emergent constraint, but an exploration of how the performance-based filtering impacts projection range compared with other sub-selection approaches. We do not conclude that the upper-end of the projection range is more credible for Europe – indeed this would not be a reasonable result as the reviewer asks, but we do think that the identified relationship between filtered ensembles and climate sensitivity highlights a tension with other potential selection approaches, such as selecting models based on global historical trends, or matching IPCC distributions of climate sensitivity. Our intention is to expose this tension for potential users of these simulations, over Europe.

These findings are complemented by a recent study that takes account of regional temperature trends which, finds that for some European areas (e.g., France), constraining the CMIP6 ensemble based on regional temperature trends, or a combination of regional and global temperature trends finds that projected summer temperature changes are shifted towards high sensitivities rather than the lower sensitivities suggested by global analyses (Qasmi and Ribes, 2022; Ribes et al., 2022). We find that the higher sensitivity models that are part of our filtered ensemble may still provide a useful projection for the European region.

 We propose to clarify in our manuscript that our result is to highlight this tension between selecting subsets based on regional performance and selecting subsets based on other criteria e.g., to represent the IPCCs plausible range of climate sensitivity.

Chen, Z. et al. (2022) 'Observationally constrained projection of Afro-Asian monsoon precipitation', Nature Communications, 13(1), p. 2552. doi: 10.1038/s41467-022-30106-z.
Jin, C., Wang, B. and Liu, J. (2020) 'Future Changes and Controlling Factors of the Eight Regional Monsoons Projected by CMIP6 Models', Journal of Climate. Boston MA, USA: American Meteorological Society, 33(21), pp. 9307–9326. doi: 10.1175/JCLI-D-20-0236.1.
Lutz, A. F. et al. (2016) 'Selecting representative climate models for climate change impact studies: an advanced envelope-based selection approach', International Journal of Climatology. John Wiley & Sons, Ltd, 36(12), pp. 3988–4005. doi: https://doi.org/10.1002/joc.4608.
Overland, J. E. et al. (2011) 'Considerations in the Selection of Global Climate Models for Regional Climate Projections: The Arctic as a Case Study', Journal of Climate. Boston MA, USA: American Meteorological Society, 24(6), pp. 1583–1597. doi: 10.1175/2010JCLI3462.1.
Qasmi, S. and Ribes, A. (2022) 'Reducing uncertainty in local temperature projections', Science Advances. American Association for the Advancement of Science, 8(41), p.

eabo6872. doi: 10.1126/sciadv.abo6872.

Ribes, A. et al. (2022) 'An updated assessment of past and future warming over France based on a regional observational constraint', Earth Syst. Dynam. Discuss., 2022(March), pp. 1–29. doi: 10.5194/esd-13-1397-2022.

Ruane, A. C. and McDermid, S. P. (2017) 'Selection of a representative subset of global climate models that captures the profile of regional changes for integrated climate impacts assessment', Earth Perspectives, 4(1), p. 1. doi: 10.1186/s40322-017-0036-4.

Whetton, P. et al. (2007) 'Assessment of the use of current climate patterns to evaluate regional enhanced greenhouse response patterns of climate models', Geophysical Research Letters. John Wiley & Sons, Ltd, 34(14). doi: https://doi.org/10.1029/2007GL030025.

2. Quantitative measures are preferred for model evaluation. Visual inspection hinders the inter-comparison of various studies to a certain degree as different people may have different judgments on "satisfactory", "unsatisfactory", and "Inadequate". I'm wondering to what extent the results will be different if the authors use objective assessment criteria only.

We understand the reviewers point that in the case of more subjective criteria, to a certain degree people may have different judgements. We have used a combination of quantitative and qualitative measures where we have found them appropriate. One point to note is that 'quantitative' is not always synonymous with 'objective' – e.g., the choice of metric and threshold for classification involves subjective judgements.

There are two main reasons for our use of qualitative measures – firstly is to account for the variety of characteristics in errors that different models display and allow us to judge their implications and significance. If we look at the climatological circulation assessment, we find that the RMSE calculated in parallel with the quantitative assessment doesn't always lead to the same classification as the visual inspection – in this case because some patterns of error are more concerning to us than others – errors in magnitude of the mean circulation (feature in broadly correct locations but with errors in magnitude) are less concerning than cases where features are incorrectly located. Visual inspection allows us to understand the characteristic of the error and consider its impact on other aspects of the model.

Figure 1 shows some examples of where bias alone and/or a RMSE threshold for windspeed would not be suitable to determine the classification of the models. BCC-CSM2-MR is classified as satisfactory for DJF circulation (Fig.1b and e). This is because although there are some errors in windspeed magnitude over western and central Europe, the pattern of large- scale circulation is reasonably well captured (as compared to ERA5 in Fig.1a). The BCC-CSM-MR model has a similar regional RMSE as BCC-ESM1 (Fig.1f), however this model is classified as unsatisfactory due to a lack of south westerly winds over the northern UK and Scandinavia (Fig,1c and f). This is

also highlighted by the negative bias in windspeed over these areas, indicating that the winds are too weak (Fig.1f). The ACCESS-ESM1-5 model (Fig 1.d and g), is also classified as unsatisfactory despite a lower regional RMSE than BCC-CSM2-MR, this is due to the wind direction being too westerly in the North Atlantic and over the UK and northern Europe. The windspeeds over Scandinavia are too weak, while the windspeed over the UK and central Europe is too strong (Fig.1g).

A quantitative metric might be designed to capture these characteristics on which our judgement is made, but this may 'miss' another error characteristic that subsequently appears in another model.

The second reason for using visual inspection is that the process of examining the fields offers us a much better understanding of model characteristics, which does not arise from summary statistics. In the study presented we have often shown quantitative metrics which were used in parallel with visual inspection.

[Figure]

*Figure 1 a) ERA5 DJF wind climatology (windspeed and direction 1995-2014), b)-d) Examples of DJF circulation patterns for model climatology. Vector arrows show absolute wind direction, contours show absolute windspeed, e)-f) As in row above but with contours showing windspeed bias compared to ERA5 (vector arrows still show absolute wind direction).*

How was the RMSE of the zonal mean track calculated? It seems that the authors calculated the zonal mean track and obtained a time series. The RMSE is calculated using the time series derived from models and observation. Please note it makes no sense by comparing the year-to-year variation of the unforced internal variability derived from AOGCMs against the observed one. In this case, the RMSE is largely determined by the phase discrepancy between simulation and observation. Please also check the use of RMSE elsewhere.

Thank you for bringing to our attention that this part of the methodology requires further clarification. The RMSE was not calculated using a time series or via consideration of each model's internal variability. This is the case for all the variables.  The zonal mean of the model mean track density from 20W-20E was taken to get a profile of storm number by latitude. Then the RMSE was calculated of the models compared to the profile obtained from ERA5. The RMSE was calculated from 25-80N. There is no timeseries element of this and it is just the RMSE of the zonal mean, model mean track density. At no point is the unforced interval variability of the models compared or used in the RMSE calculations. We will clarify this in the paper's text.

Other comments:

Section 2: It is not clear to me how the CMIP6 models are grouped into classifications. Please clarify how the quantitative and qualitative measures were used and what is the threshold of quantitative measures to group the models. I suggest the authors introduce the "criteria" first and explain the classification definitions based on the criteria.

Models were classified for individual criteria and not grouped into an overall classification (figures 4 and 5 in the manuscript).  Models were then sub-selected based on whether they had any red flags (inadequate) and the percentage of orange (unsatisfactory) flags.  This is presented as only one example of how the assessment can be used to sub-select models. An alternative approach, for example, would be to only remove models with an inadequate flag. Thank you for this suggestion we will improve the clarity of the main text.

L64: "processed based" -> "process-based"

L70: How the regional processes are linked to future changes?

L137: "process base" -> "process-based", "does not use and regional or global warming trends"->"does not use regional or global warming trends". Please carefully

read throughout the manuscript and correct the typos or grammar mistakes. E.g. L202 ...

Thank you for noting these errors, these will be corrected, and the final manuscript will be proofread.

L217: What is the temporal resolution of the dataset, monthly mean or daily mean? Which CMIP6 experiment was used for the baseline period? Both the baseline and future periods are only 20 years. The climatological means averaged over 20 years may still contain internal climate variability, e.g., AMO or PDO, which may affect the evaluation and selection of the models to a certain extent.

Thank you for highlighting that this is not clear, we use monthly datasets and the historical experiment for the baseline. We have selected the time periods used in the assessment to align with the European Projections Project (e.g., Brunner et al., 2020).

Brunner, L. et al. (2020) 'Comparing Methods to Constrain Future European Climate Projections Using a Consistent Framework', Journal of Climate, 33(20), pp. 8671–8692. doi: 10.1175/JCLI-D-19-0953.1

L225: Please clarify what reanalysis and observational data were used in this study.

ERA5 was the reanalysis data used for the assessment criteria. The exception is for precipitation where EOBS data was used. This will be clarified in the text.

L254-255: How the circulation pattern is measured? Is the RMSE calculated using two wind speed fields or an RMS vector error between two vector fields? If the RMSE is calculated with wind speed, it does not reflect the errors in wind direction. Instead, the RMSE for vector field can reflect both errors in wind speed and wind direction. Therefore, I suggest the authors use the latter one. Similarly, the difference in wind speed illustrated in Fig. 1 can only describe the errors in wind speed. The same wind speed does not mean the same wind direction. The authors may consider using a vector difference between the model and ERA5. The magnitude of vector difference takes both differences in wind speed and wind direction into account.

The wind speed was used as a measure of the magnitude of error, while the circulation pattern of wind direction and magnitude was assessed visually. Thank you for this suggestion, it may be interesting to use the vector error in addition to the windspeed and see if this is a better indicator of errors in the circulation pattern.

Xu et al, 2016: A diagram for evaluating multiple aspects of model performance in simulating vector fields. Geosci. Model Dev., 9, 4365–4380

L270: Please explain how the "track density" is defined. Please use the degree symbol "°" to represent latitude and longitude here and elsewhere.

The track density is calculated using an objective cyclone tracking and identification method based on 850 hPa relative vorticity (Hodges, 1994, 1995). The method and data are the same used in Priestley et al. (2020). This will be clarified in the manuscript.

Hodges, K. I., 1994: A general method for tracking analysis and its application to meteorological data. Mon. Wea. Rev., 122, 2573–2586, https://doi.org/10.1175/1520-0493(1994)122,2573: AGMFTA.2.0.CO;2.
Hodges, K. I., 1995: Feature tracking on the unit sphere. Mon. Wea. Rev., 123, 3458–3465, https://doi.org/10.1175/1520-0493(1995)123,3458: FTOTUS.2.0.CO;2.
Priestley, M. D. K. et al. (2020) 'An Overview of the Extratropical Storm Tracks in CMIP6 Historical Simulations', Journal of Climate. Boston MA, USA: American Meteorological Society, 33(15), pp. 6315–6343. doi: 10.1175/JCLI-D-19-0928.1

L321: "depending to on" -> "depending on"

L334: "with with" -> "with"

L343-345: How about the range of other quantities, e.g. precipitation and storm track density?

The authors agree that it would be interesting to investigate other variables, it would extent the scope and length of the existing paper considerably to consider projections from the filtered ensemble for all the criteria that have been assessed. This is something that the authors are interested in exploring further and in a more thoroughly in a second follow up paper.

L362: Please clarify what numerical score was given for each group of models.

This information can be added to the manuscript in the supplementary info. Each model was scored individually. Models were classified for individual criteria and not given an overall classification. Models were then sub-selected based on whether they had any red flags (inadequate) and the percentage of orange (unsatisfactory) flags. This is presented as one example of how the assessment can be used to sub-select models. Model sub-selection is always subjective to some extent and the approach will depend on the application.

L644: "35°N-75°" -> "35°N-75°N"

Fig. S4: What does the "??" refer to in the figure caption?

This is a typo, it refers to table 2 in the main manuscript, thank you for noting this, it will be corrected.

---

## Author Response (AR1)

**Response reviewer 1**

Peer review of "Performance based sub-selection of CMIP6 models for impact assessments in Europe" by Palmer et al. (ESD).

This paper presents a performance assessment of CMIP6 simulations for Europe and selects a subset of models for regional climate impact studies. The performance criteria include large-scale processes such as storm tracks, circulation patterns, and temperature biases. The selection of models is based primarily on subjective assignment of each model into three categories for each performance criterion. The authors highlight that there is a strong tendency for the models with high regional performance to have higher global climate sensitivity. While the causes of this relationship is left for future investigation, the authors note that this relationship creates a tension between selecting for high regional performance and selecting an ensemble consistent with observational constraints on global ECS.

This paper is thoughtful and well executed. It will be useful for European climate impact assessments, and also as a template/benchmark for performance assessments in other regions. While the paper is acceptable with minor technical corrections, I have added some optional suggestions for improvement. The most important of these suggestions is for an assessment of the role of internal variability in the performance evaluation.

*We thank the reviewer for their overall positive and very constructive response. Along with their helpful suggestions for improving the manuscript. Our initial response is given below.*

**Corrections required:**

There are many spelling and grammar mistakes. I noted typos in lines 10, 52, 69, 82, 114, 137, 188, 202, 211, 238, 245, 279, 319, 330, 426, 432, 436, 444, 464, 467, and 471 and in the spelling of "conceptulization."

Table 1. ACCESS-CM2 Is missing from the left column. Also, since the right column is a subset of the left, couldn't this table be replaced with a (less space-consuming) list, with selected models highlighted in bold?

Table 2. The selected model in each cluster needs to be identified. This info isn't available from figure 7 or anywhere else in the main text.

We thank the reviewer for noting these errors. The final manuscript has been proof-read, and the errors noted above corrected.

The oversight of ACCESS-CM2 has been corrected and Table 1 has also been replaced with a list as suggested here. See lines 446-449:

*Once all the models in Fig. 5 that have a red flag for the large-scale criteria are removed the following models remain in the sub-selection; ACCESS-CM2, BCC-CSM2-MR, CESM2, CESM2-WACCM, CNRM-CM6-1, CNRM-CM6-1-HR, CNRM435 ESM2-1, EC-Earth3. EC-Earth3-Veg, GFDL-CM4, GFDL-ESM4, HadGEM3-GC31-LL, HadGEM-GC31-MM, MPI-ESM1- 2-HR, MRI-ESM2-0, KACE-1-0-G, TaiEMS1, UKESM1-0-LL*

The selected model from each cluster has been identified in table 2 in bold. All the models in figure 7 have been identified by numbering the points in the supplementary material (Fig. S4).

**Suggestions for improvement (optional):**

Models are evaluated on the basis of a single realization each. To what extent does internal variability affect the assessments? The paper would be more solid if it included an analysis of the robustness of the performance criteria to multiple realizations of at least one model.

We agree with the reviewer that the assessment would be more robust with an understanding of the importance of internal variability and acknowledge that for individual models and criteria is likely that the assessment classifications that we use will not be identical across all realisations (this may especially be the case were models are close to classification thresholds. While we have not been able to address this in full, this has been given some further consideration in the manuscript.

*See lines 275 – 292:*

*We use only the first realisation for each of the models in this assessment and assume that this is generally representative of the model performance. We acknowledge however that there may be a role for internal variability that pushes a model across assessment classifications. The largest uncertainty due to internal variability of the diagnostics we use is likely to be from the historical trends (which are not part of the assessment but used in an illustrative capacity). Brunner et al. (2020) found that for the global case the spread in the temperature trend fields between ensembles members of one model can be in the same order of magnitude as the spread across the multi-model ensemble. For the temperature climatology, in turn, the spread between ensemble members of the same model is typically less than 10% of the multi-model spread. This gives some indication that we can expect there to be relatively low variation in the performance of the models across the climatology for*

*temperature based on which member is used. For the AMOC, which is a significant contributor to regional and global climate variability, Menary et al. (2020) noted that links to North Atlantic SSTs were sensitive to the removal (or not) of forced variability, but individual model realisations were not systematically different. 9 A case study is made to assess the role of internal variability for large-scale circulation (in which we may expect larger variability across ensemble members, than for the temperature climatology) in the CanESM5 model across all 25 realisations. This can be viewed in the supplementary information (Figs. S5 and S6). This context suggests that the analysis presented in this paper, based on the first ensemble member, likely provides an indicative picture typical of the response across any wider initial condition ensemble. However, future assessments may want to look for individual ensemble members which may show weaker manifestations of particular biases, particularly where a model lies close to classification boundaries.*

[Figure]

*Figure S6 DJF circulation (850hPa) classifications for the CanESM5 realisations. Top panel shows ERA5 climatology. Windspeed and direction are shown as a 20 year mean 1995 − 2014. Arrows show wind direction (absolute, scaled by windspeed) for climatology across all panels The shading for the 3 panels shows the difference in windspeed between the realisation and ERA climatology.*

[Figure]

*Figure S6 JJA circulation (850hPa) classifications for the CanESM5 realisations. Top panel shows ERA5 climatology. Windspeed and direction are shown as a 20 year mean 1995 − 2014. Arrows show wind direction (absolute, scaled by windspeed) for climatology across all panels. The shading for the 3 panels shows the difference in windspeed between the realisation and ERA climatology*

Further text has been added in the supplementary information see text lines 1 to 9 :

*From Fig S5 and S6 it can be seen that the pattern of errors remains strikingly similar across the ensemble realisations for both DJF and JJA. There is however variability in the severity of the errors. For example, for DJF the large-scale circulation for CanESM5 was found to be 'Inadequate', based on the first ensemble member. While many of the ensemble members have larger errors that the first member (e.g. r18i1p1f1, r22i1p1f1), there is one realisation (r61p1f1) that would likely to qualify as 'Unsatisfactory'. The patter of errors for JJA is again very similar across the realisations, but there is some variation in the magnitude of the errors. From this we conclude that there may be instances where the varibilty across realisations means that a model that has been classified as 'Inadequate' may have some realisations where the performance of the model might be acceptable. It is also likely be the case that where a model has been classified as 'Unsatisfactory', some individual realisations could be considered 'Inadequate'.*

This paper's strength is in the process evaluations, which will be a useful reference for analysts creating bespoke ensembles. The 3x3 matrix of examples of models in the three subjective categories is a nice way of presenting the results in the main paper and the appendix. however, many analysts would benefit from a supplementary section showing the maps for the full set of assessed models, so they can make their own subjective assessments and better understand figures 4 and 5.

We agree with the reviewer that it would be beneficial to users to make the maps for the full set of assessed models available. An accessible github has been created, line 630:

*A version of the assessment figures used in this paper is available on github https://github.com/tepmo42/cmip6_european_assessment as a full spreadsheet of all available assessments (for Europe) carried out for CMIP6 models to date.*

This includes the maps used in the assessment for temperature, large-scale, blocking frequency and plots for the precipitation annual cycle. This repository will enable this to potentially be maintained as a living document, that can be added to as more models or diagnostics become available.

The finding that many of the high-skill models are outside the IPCC assessed ECS range is interesting and important. However, this tension between regional skill and global climate sensitivity seems somewhat overstated. There are a couple of solutions that partially resolve this tension. First, there is the option of presenting analyses relative to global warming levels instead of time, as widely practiced in the literature and advocated by Hausfather et al. (2022). While the GWL approach doesn't fully resolve the tension (time does matter to many studies), it warrants some discussion here. Indeed, the results of this paper add further weight to the importance of the GWL approach. Second, the IPCC's very likely ECS range is a more inclusive and defensible (66% is a high bar, given the observational uncertainties on the upper tail of ECS) criterion that would only exclude three independent models (CanESM, UKESM/HadGEM, and CESM2). Discussion of these nuances would give more direction to the reader in the face of the tension that this paper highlights.

The very likely IPCC range for ECS is shown in grey on Figure 7b), The suggestion of selecting models from this range has also been added to manuscript.

We also agree with the reviewer that the tension between the IPCC assessed climate sensitivity range, and the regional skill of the models is not an issue if the GWL method is a suitable approach. Some discussion of this has been added to the manuscript. Lines 502 – 506:

*It may be appropriate to select only the better performing the models from within the very likely IPCC range for ECS, or to retain just one of the models above this range to account for a higher impact scenario. It may also be appropriate to select models that are 'marginal' from the lower part of the IPCC very likely range. Alternative using an approach that considers regional impacts using Global Warming Levels could be applied to the sub-set, this is discussed further in section 5.*

And lines: 581-583:

*This apparent tension is likely to be less evident where Global Warming Levels are instead adopted. Using this approach, when adopting a selection based on climate performance, such as presented here, would enable a broader set of "adequate" realisations to be explored.*

There are however cases where the GWL method is not suitable, such as, where the distribution of the ensemble is used as a measure of likelihood. As shown in our results, the distribution of the filtered models with greater regional skill is skewed towards higher climate sensitivity. In this case the tension between the regional skill and climate sensitivity then becomes relevant. It is particularly important where assessments are made by a risk adverse user, that is interested in a high impact, low likely hood (but plausible) temperature change within a given time frame (e.g., 2030 or 2040).

The completeness of scenario experiments by each model is an important consideration in ensemble selection that doesn't receive any attention here. For example, HadGEM3-GC3.1 provides only one simulation of SSP126 and no simulations of SSP370 (https://pcmdi.llnl.gov/CMIP6/ArchiveStatistics/esgf_data_holdings/ScenarioMIP/index.html), and as a result may not be viable for some study designs. The paper could benefit from some documentation and/or discussion of this and other practical considerations that will affect the utility of the recommended ensemble

We agree that the completeness of the scenario experiments is likely to be a consideration for users. The focus of the paper is on the process-based assessment, rather than attempting to address some of the wider potential considerations for selecting representative models, for downscaling and impact assessments. However, we agree that some relevant links to the documentation would be useful and have added this to the supplementary information in the caption for table S1.

*Information regarding the available scenario experiments for the CMIP6 models is available at:*
*https://pcmdi.llnl.gov/CMIP6/ArchiveStatistics/esgf_data_holdings/ScenarioMIP/index.html*

The exclusion of UKESM1 based on orange flags comes across as a bit haphazard and arbitrary, especially given that analysis of storm track performance is not available for this model. While I noted the discussion on the confluence of reasons for excluding UKESM1, the paper would benefit from a more systematic documentation of the interaction of criteria leading to model exclusion. Perhaps also there is a role for a "marginal" category of models for which exclusion wasn't clear-cut.

This is useful feedback. We agree that the decision to remove models due to a certain percentage of orange flags is somewhat arbitrary. The decision as to how many orange flags warrants the removal of a model is somewhat arbitrary. In the revised manuscript we use a more consistent approach that refers to the original classification definitions. Here we only define a classification of 'Inadequate', to warrant the removal of a model. Section 4.2 has been revised to reflect this and we explore instead the difference in the sub-set for large-scale criteria only, and a second sub-set that includes the regional criteria. An additional Figure (Fig.6) has been added that shows the change in range for the large-scale filtering (this includes the UKEMS1 model). We acknowledge that the 'Inadequate' flag in a local region may not always warrant the removal of a model. This is particularly relevant to the UKESM1 model, which performs relatively well against the most the criteria, except for a large winter bias in Northern Europe. See section 4.2, in particular lines 436-475:

[revised manuscript text omitted]

**Minor comments:**

Line 225. Some more detail on the reanalysis/observational data would be helpful –

The following has been added to the methodology lines 266-270:

*The ERA5 reanalysis data and E-OBS gridded observational dataset (to evaluate the precipitation annual cycle) were used to assess the model error. Monthly mean data is used for the assessment with the exception of the blocking frequency analysis which uses daily data fields. Details of how these assessments have been carried out for each of the criteria, are given in the appendices*

Lines 427-8. "The retention of higher sensitivity models is an emergent consequence of assessment of skill at reproducing regional processes." This wording implies some functional relationship between regional skill and model sensitivity that hasn't been established (as duly noted in the conclusion). Simpler wording would reduce the chance of misinterpretation by the reader.

This has been reworded lines 570-572:

*The retention of higher sensitivity models is due to more of the higher sensitivity models demonstrating a greater skill for reproducing regional processes.*

Lines 459-60. Shiogama (2021) excluded models based on a criterion of high recent warming relative to observations, rather than based on ECS or TCR as implied here. Mahony (2022) (DOI:10.1002/joc.7566) would be a more direct example of ensemble selection based on the IPCC assessed ECS range.

Thank you for this suggestion this reference has been updated in the manuscript, now at lines: 576- 578

*Using the IPCC AR6 likely range for ECS (and or TCR, Hausfather et al. (2022)) has also been suggested as an approach to model screening for the CMIP6 ensemble. Other regional sub-selection studies for CMIP6 have eliminated models with high global sensitivity (Mahony et al., 2022)*

**Reviewer 2 response**

Recommendation: Major revision

The authors assessed CMIP6 models in terms of their performance and diversity in simulating several variables, e.g., temperature, precipitation, and circulation, over Europe. Based on the assessment, they created sub-sets of CMIP6 models, which can be used for downscaling or impacts assessments. The approach can also be applied to other regions of the world. The topic is important and falls within the scope of the journal. The manuscript is generally well written. My major concern includes: the assessment of CMIP6 models did not well consider the link between the model's ability to simulate historical climate and future climate change. The assessments are overly dependent on subjective assessment criteria. Detailed comments are laid out below.

We thank the reviewer for this overall positive and constructive response.

Major comments:

1. No link was established in terms of the model's ability to simulate the historical climate and the projected changes. Thus, the models that can better reproduce historical climate may not necessarily generate a more reliable projection of future climate. After excluding the least realistic models, the filtered CMIP6 models show higher sensitivity. Is the result reasonable?

It is correct that we do not attempt to explicitly link baseline performance to the credibility of future projections. What we do suggest, is that there are a number of issues around using climate model projections from models which do not behave realistically in terms of key large scale regional climate characteristics in the baseline climate. We have added some further discussion of this to the introduction in lines: 31-55

*Whetton et al., (2007) evaluate the link between model performance in the historical period and model performance for future projections by investigating the model similarity in patterns of the current climate and the inter-model similarity in regional patterns in response to CO2 forcing. They find that similarity in current climate regional patterns of temperature, precipitation and MSLP from GCMs is related to similarity in the patterns of change of these variables in the models.*

*In addition, while global temperature biases in the historical record are not correlated with future projected warming (e.g., Flato et al., 2013). This is not the case regionally for Europe, where biases in the summer temperatures have been found to be important for constraining future projections (Selten et al., 2020). In addition projections of the Artic sea ice extent have also been linked to historical temperature biases (Knutti et al., 2017) .*

*An increasing body of literature does link short comings in the ability of a model to realistically represent an observed baseline to being an indicator that the models' future projections are less reliable (e.g., Whetton et al., 2007; Overland et al., 2011; Lutz et al., 2016; Jin et al., 2020; Chen et al., 2022; Ruane and McDermid, 2017).*

*Regional model sub-selection is guided by a range of choices and there is always an element of subjectivity in terms of how the criteria are determined. For example, if a model performs well for a particular target variable, but then performs poorly in another season, variable, or location, this indicates that the regional climate processes are suspect (Whetton et al., 2007; Overland et al., 2011).*

*To assess the model performance in terms of the regional climate processes, we firstly identify the key drivers of the European climate as our criteria. We then use these to assess the performance of the CMIP6 models across a range of variables. The approach that we take one of elimination rather than selection and we do not recommend any individual model. Rather in our 2 examples of approach to sub-selection, we examine the impact on the projection range from the elimination of the models that perform relatively poorly in these key criteria.*

Having identified models that we consider particularly unrealistic to arrive at a filtered subset, we then explore what that means for the range of future projections. We find that the better-performing filtered subset happens to contain a higher proportion of higher sensitivity models. This study is not intended to present an emergent constraint, but an exploration of how the performance-based filtering impacts projection range compared with other sub-selection approaches. We do not conclude that the upper-end of the projection range is more credible for Europe – indeed this would not be a reasonable result as the reviewer asks, but we do think that the identified relationship between filtered ensembles and climate sensitivity highlights a tension with other potential selection approaches, such as selecting models based on global historical trends, or matching IPCC distributions of climate sensitivity. Our intention is to expose this tension for potential users of these simulations, over Europe.

Our findings are complemented by a recent study that takes account of regional temperature trends which, finds that for some European areas (e.g., France), constraining the CMIP6 ensemble based on regional temperature trends, or a combination of regional and global temperature trends finds that projected summer temperature changes are shifted towards high sensitivities rather than the lower sensitivities suggested by global analyses (Qasmi and Ribes, 2022; Ribes et al., 2022). We find that the higher sensitivity models that are part of our filtered ensemble may still provide a useful projection for the European region.

We have clarified this in the text with some further discussion regarding how the assessment might want to be used alongside the ECS or potentially considering performance against the global historical trend. We have added some further discussion and clarified this in lines 499-506

*'Our result does not include any consideration of climate sensitivity and while these models are identified here as performing relatively well in a process-based assessment, the sub-set temperature range shown in Fig. 7 should not be viewed as a constraint that gives a more accurate projected range for Europe. Here we only highlight that more of the models that perform well in terms of regional physical processes have a higher climate sensitivity. It may be appropriate to select only the better performing the models from within the very likely IPCC range for ECS, or to retain just one of the models above this range to account for a higher impact scenario. It may also be appropriate to select models that are 'marginal' from the lower part of the IPCC very likely range. Alternative using an approach that considers regional impacts using Global Warming Levels could be applied to the sub-set, this is discussed further in section 5.'*

2. Quantitative measures are preferred for model evaluation. Visual inspection hinders the inter-comparison of various studies to a certain degree as different people may have different judgments on "satisfactory", "unsatisfactory", and "Inadequate". I'm wondering to what extent the results will be different if the authors use objective assessment criteria only.

We understand and agree with the reviewer's concern that some of the classifications that are based more on a qualitative assessment and to some degree different people may have different judgements. We further agree that it is important that it does results in objectively worse models with clearly larger errors being retained, while a model with objectively smaller errors is removed.

One point to note is that 'quantitative' is not always synonymous with 'objective' – e.g., the choice of metric and threshold for classification involves subjective judgements.

The reason for our use of qualitative measures is to account for the variety of characteristics in errors that different models display and allow us to judge their implications and significance. Some explanation for our additional visual inspection of the error fields (in addition to the consideration of quantitative measures) is added at lines 228-233:

*An additional qualitative element to the assessment can add value by interpreting how these errors impact on the overall performance of the model for the regional climate and helps to inform the question of why these errors may cause a model projection to be less reliable.*

*A mix of quantitative (RMSE, bias, variance, correlation) and qualitative (e.g., inspection of circulation wind patterns) have been used and the models graded for each criterion using a coloured flag system. The reason for our use of qualitative measures is to account for the variety of characteristics in errors that different models display and allow us to judge their implications and significance. Visual inspection allows us to understand the characteristic of the error and consider its impact on other aspects of the model.*

Many of the classifications are chosen based on the range of RMSE or a combination or correlation, bias and RMSE. For example, the red, 'Inadequate' category for storm tracks was chosen based on the 85th percentile of the RMSE. The models that fell into this range of errors were found to unable to capture the trimodal pattern as well as having large errors in magnitude. The blocking frequency classifications were also sorted by a k-clustering algorithm based on a combination of bias, RMSE and correlation with the reanalysis data.

To aid transparency our qualitative assessment has now been presented alongside quantitative scores for the model RMSE. To address the reviewer's question regarding any difference in the results from using a simple objective measure of error, we have added a table of RMSE of the model errors for the area which each of the criteria were assessed over for the large-scale criteria (Fig 1).

[Figure]

*Figure 1 . Summary of RMSE values for the large-scale assessment criteria. The colour scale is determined by the ratio of the model RMSE. RMSE values are absolute, the mean score is the average of the relative error (normalised by the ensemble mean) across each of the criteria.*

In addition, some of the more qualitative assessments have been reviewed. A simpler quantitative RMSE in two areas has now been used to assess the North Atlantic Sea Surface Temperature (NA SST) See section A2 in the appendices in particular lines 662 - 674:

[revised manuscript text omitted]

Overall, in our assessment's quantitative measures along with visual inspection of the model fields have been used to assess which category the models should fit into.

The assessment of the large-scale circulation was also reviewed using the wind vector errors as suggested by the reviewer. We thank the reviewer for this useful suggestion, please see these revisions detailed under the relevant comment.

**I'm wondering to what extent the results will be different if the authors use objective assessment criteria only.**

This is an important question and one that we have ensured is addressed in the revised paper with a comparison of the sub-set of models from the qualitative assessment with a simple summary of the RMSE for the large-scale criteria (Fig 1)
.
The thresholds for the categories were chosen based largely on the relative performance of the models. Therefore, it is the poorest models relative to the rest of the ensemble that have been removed. This has been confirmed by comparison with Fig.1.  We add some discussion of this in lines 449-455:

*If we look at the scores for the large-scale criteria (all categories in Fig 1, excluding regional temperature), it can be seen that the excluded models include all those with a RMSE more than 1.5 times the ensemble mean in at least one of the large scale categories. It is also the case that for the retained models that the RMSE does not exceed 1.5 times the multi-model ensemble mean for any large-scale category. The retained models also perform better than, or at least equal to the ensemble mean across all the categories. This indicates that in our application of the assessment objectively poorer models have been removed (in terms of large-scale performance) and those with objectively smaller errors have been retained.*

And lines 472-475 :

*Referring to Fig. 1, we can confirm that our excluded models include only those with relative 460 large RMSE (1.5 times the ensemble mean) in at least one of the criteria. Also, that the eliminated models on average across the criteria have a relative error at least equal to or larger than the ensemble mean (Fig. 1. Therefore, it is again the case that the qualitative assessment has removed the models with objectively larger errors in the key criteria.*

How was the RMSE of the zonal mean track calculated? It seems that the authors calculated the zonal mean track and obtained a time series. The RMSE is calculated

using the time series derived from models and observation. Please note it makes no sense by comparing the year-to-year variation of the unforced internal variability derived from AOGCMs against the observed one. In this case, the RMSE is largely determined by the phase discrepancy between simulation and observation. Please also check the use of RMSE elsewhere.

Thank you for bringing to our attention that this part of the methodology requires further clarification. The RMSE was not calculated using a time series or via consideration of each model's internal variability. This is the case for all the variables. The zonal mean of the model mean track density from 20W-20E was taken to get a profile of storm number by latitude. Then the RMSE was calculated of the models compared to the profile obtained from ERA5. The RMSE was calculated from 25-80N. There is no timeseries element of this and it is just the RMSE of the zonal mean, model mean track density. At no point is the unforced interval variability of the models compared or used in the RMSE calculations. See lines *371-373:*

*'The zonal mean of the model mean track density from 20°W-20°E was taken to get a profile of storm number by latitude. Then the RMSE was calculated of the models compared to the profile obtained from ERA5. The RMSE was calculated from 25-80°N'.*

Other comments:

Section 2: It is not clear to me how the CMIP6 models are grouped into classifications. Please clarify how the quantitative and qualitative measures were used and what is the threshold of quantitative measures to group the models. I suggest the authors introduce the "criteria" first and explain the classification definitions based on the criteria.

The order has now been changed in the text, the criteria are now introduced first followed by the classification definitions. These definitions have been used in the qualitative assessment to choose suitable quantitative thresholds for each of the criteria.

How the thresholds are selected for each of the criteria is explained in the individual sections. Most of these are in the appendices, examples for large-scale and storm tracks are in sections 3.2.1 and 3.2.2. There are a few cases where a models lie on the borderline for the quantitative thresholds and a 'hard' threshold may not always be appropriate, these cases have been all been discussed individually in the relevant sections.

To make the assessment more transparent we have included (see figure 1, above), the RMSE for each of the large-scale assessment criteria, this can be compared to the assessment flags for each of the models. The plots used in the assessment for all the models are also available at (see line 630):

*A version of the assessment figures used in this paper is available on github https://github.com/tepmo42/cmip6_european_assessment as a full spreadsheet of all available assessments (for Europe) carried out for CMIP6 models to date.*

Models were classified for individual criteria and not grouped into an overall classification (see figure 5 in the manuscript). Models were then sub-selected based on whether they had any red flags ('Inadequate') in section 4.2. This is presented as only one example of how the assessment can be used to sub-select models.

L64: "processed based" -> "process-based"

L70: How the regional processes are linked to future changes?

We have added further discussion in the manuscript regarding the link to future changes, please also see response to main point 1. see lines 31-49.

*Whetton et al., (2007) evaluate the link between model performance in the historical period and model performance for future projections by investigating the model similarity in patterns of the current climate and the inter-model similarity in regional patterns in response to CO2 forcing. They find that similarity in current climate regional patterns of temperature, precipitation and MSLP from GCMs is related to similarity in the patterns of change of these variables in the models.*

*In addition, while global temperature biases in the historical record are not correlated with future projected warming (e.g., Flato et al., 2013). This is not the case regionally for Europe, where biases in the summer temperatures have been found to be important for constraining future projections (Selten et al., 2020). In addition projections of the Artic sea ice extent have also been linked to historical temperature biases (Knutti et al., 2017) .*

*An increasing body of literature does link short comings in the ability of a model to realistically represent an observed baseline to being an indicator that the models' future projections are less reliable (e.g., Whetton et al., 2007; Overland et al., 2011; Lutz et al., 2016; Jin et al., 2020; Chen et al., 2022; Ruane and McDermid, 2017).*

*Regional model sub-selection is guided by a range of choices and there is always an element of subjectivity in terms of how the criteria are determined. For example, if a model performs well for a particular target variable, but then performs poorly in another season, variable, or location, this indicates that the regional climate processes are suspect (Whetton et al., 2007; Overland et al., 2011).*

L137: "process base" -> "process-based", "does not use and regional or global warming trends"->"does not use regional or global warming trends". Please carefully read throughout the manuscript and correct the typos or grammar mistakes. E.g. L202 ...

Thank you for noting these errors, these will be corrected.

L217: What is the temporal resolution of the dataset, monthly mean or daily mean? Which CMIP6 experiment was used for the baseline period? Both the baseline and future periods are only 20 years. The climatological means averaged over 20 years may still contain internal climate variability, e.g., AMO or PDO, which may affect the evaluation and selection of the models to a certain extent.

The following has been added for clarification. Line 266

*Monthly mean data is used for the assessment with the exception of the blocking frequency analysis, which uses daily data fields.*

The historical time periods used in each of the assessments is detailed in the relevant section. Some of these are in the appendices. While we generally use the 1995-2014 historical period for consistency with other studies of European Climate Projections (EUCP), where we incorporate analyses already in the literature (e.g., Priestly et al., 2020) the historical periods for these are used. This includes longer/different periods for key atmospheric assessments, for storm tracks this used the 1979-2000 period and for blocking frequency the assessment is from 1961-2003. These are detailed in the relevant sections.

L225: Please clarify what reanalysis and observational data were used in this study.

The following text has been added to the methodology line 266-268:

*The ERA5 reanalysis data and E-OBS gridded observational dataset (to evaluate the precipitation annual cycle) were used to assess the model error. Monthly mean data is used for the assessment with the exception of the blocking frequency analysis, which uses daily data fields. Details of how these assessments have been carried out for each of the criteria, are given in the appendices.*

L254-255: How the circulation pattern is measured? Is the RMSE calculated using two wind speed fields or an RMS vector error between two vector fields? If the RMSE is calculated with wind speed, it does not reflect the errors in wind direction. Instead, the RMSE for vector field can reflect both errors in wind speed and wind direction. Therefore, I suggest the authors use the latter one. Similarly, the difference in wind speed illustrated in Fig. 1 can only describe the errors in wind speed. The same wind speed does not mean the same wind direction. The authors may consider using a vector difference between the model and ERA5. The magnitude of vector difference takes both differences in wind speed and wind direction into account.

Xu et al, 2016: A diagram for evaluating multiple aspects of model performance in simulating vector fields. Geosci. Model Dev., 9, 4365–4380

The reviewer is correct that we used the RMSE wind speed error and does not reflect the errors in wind direction. We used a visual comparison in the original submission to assess the windspeed. This is one reason why the overall RMSE and assessments are not always aligned in the original manuscript.

In the revised manuscript we have used the vector difference to assess the circulation patters quantitatively as suggested (see figure 1). We agree that this provides a better quantitative assessment of the circulation patterns, and these errors now align closely with our qualitative assessment categories. However, we still find that a qualitative to understanding of the errors still adds value to our assessment. From the RMSE values alone we can rank the performance of the models, but it is not clear to what degree the different errors affect the representation of the circulation. Any exceptions or models borderline in the category based on the vector RMSE have been discussed in the text. See revised section 3.2.1 lines 307-368:

[revised manuscript text omitted]

Deciding a threshold for where models should be eliminated in sub-selection is always subjective to a degree whether quantitative or qualitative measures are used. We have aimed to add value by using a qualitative approach to indicate where the errors result in the model being unable to realistically represent the regional climate. Our criteria our designed to complimentary in this respect and used to create a summary of the model from the combined process-based performance indicators.

L270: Please explain how the "track density" is defined. Please use the degree symbol "°" to represent latitude and longitude here and elsewhere.

Some further information has been added at lines: 370-371

*'The track density is calculated using an objective cyclone tracking and identification method based on 850 hPa relative vorticity (Hodges, 1994, 1995). The method and data are the same used in Priestley et al. (2020).'*

L321: "depending to on" -> "depending on"

L334: "with with" -> "with"

Thank you for noting these errors.

L343-345: How about the range of other quantities, e.g. precipitation and storm track density?

The authors agree that it would be interesting to investigate other variables, it would extent the scope and length of the existing paper considerably to consider projections from the filtered ensemble for all the criteria that have been assessed. This is something that the authors are interested in exploring further and in a more thoroughly in a second follow up paper.

L362: Please clarify what numerical score was given for each group of models.

This has been clarified in the text, see lines (493-497)

*A regional consolidated performance index was created by giving the satisfactory (white), unsatisfactory (orange) and inadequate (red) flags a numerical score of 1 for 'Satisfactory', 2 for 'Unsatisfactory' and 3 for 'Inadequate. The overall score for each model was then averaged by the total number of assessed criteria, to give an indication of how the model performed overall. Many of the models that performed well for the process-based criteria do not fall within the IPCC AR6 likely range for equilibrium climate sensitivity (ECS) (Forster et al., 2021) (Fig. 7b)*

This information for the scores of each model is available in a github repository , line 360:

*A version of the assessment figures used in this paper is available on github https://github.com/tepmo42/cmip6_european_assessment as a full spreadsheet of all available assessments (for Europe) carried out for CMIP6 models to date.*

L644: "35°N-75°" -> "35°N-75°N"

Thank you for noting this is has been corrected.

Fig. S4: What does the "??" refer to in the figure caption?

This is a typo, it refers to table 2 in the main manuscript, thank you for noting this, it will be corrected.